# Quantum chaos in 2D gravity

Alexander Altland[1][*], Boris Post[2][†], Julian Sonner[3][‡],
Jeremy van der Heijden[2][∘] and Erik Verlinde[2][§]

**1** Institut für Theoretische Physik, Universität zu Köln,
Zülpicher Str. 77, 50937 Köln, Germany
**2** Institute for Theoretical Physics, University of Amsterdam, Science Park 904,
Postbus 94485, 1090 GL Amsterdam, The Netherlands
**3** Department of Theoretical Physics, University of Geneva,
24 quai Ernest-Ansermet, 1211 Genève 4, Suisse

[*] alexal@thp.uni-koeln.de , [†] b.p.post@uva.nl , [‡] julian.sonner@unige.ch
[∘] j.j.vanderheijden2@uva.nl , [§] e.p.verlinde@uva.nl

## Abstract

We present a quantitative and fully non-perturbative description of the ergodic phase of quantum chaos in the setting of two-dimensional gravity. To this end we describe the doubly non-perturbative completion of semiclassical 2D gravity in terms of its associated universe field theory. The guiding principle of our analysis is a flavor-matrix theory (fMT) description of the ergodic phase of holographic gravity, which exhibits $U(n|n)$ causal symmetry breaking and restoration. JT gravity and its 2D-gravity cousins alone do not realize an action principle with causal symmetry, however we demonstrate that their *universe field theory*, the Kodaira-Spencer (KS) theory of gravity, does. After directly deriving the fMT from brane-antibrane correlators in KS theory, we show that causal symmetry breaking and restoration can be understood geometrically in terms of different (topological) D-brane vacua. We interpret our results in terms of an open-closed string duality between holomorphic Chern-Simons theory and its closed-string equivalent, the KS theory of gravity. Emphasis will be put on relating these geometric principles to the characteristic spectral correlations of the quantum ergodic phase.

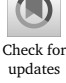

## 1 Introduction

In recent years, there has been a renewed interest in two-dimensional quantum gravity, most notably JT gravity [1–5], its supersymmetric cousins [6–8], and more general dilaton gravity theories [9–11]. In particular, the focus has been on understanding these toy models of quantum gravity at a fully non-perturbative level. Much research has concentrated on the proposed completions as double-scaled matrix models, whose universal features capture hallmarks of an underlying chaotic microscopic theory, such as the plateau in the spectral form factor [12,13]. However, an alternative approach proposed in [14,15] treats the two-dimensional JT universe as the worldsheet of a closed topological string, whose splitting and joining is described by a field theory in target space. The diagrammatic expansion of this simple interacting 2d CFT – dubbed a 'universe field theory', after [16] – corresponds to the genus expansion of the gravitational path integral, while non-perturbative information can be accessed by allowing JT strings to end on D-branes. Besides conceptually clarifying the matrix model origins of JT gravity, it also offers the technical advantage of working directly in the double-scaling limit.

In this article we use our universe field theory to connect JT gravity to a hallmark in the field of quantum chaos, namely the 'supersymmetry method', as pioneered by Efetov [17] and recently applied to holography in [18, 19]. Given some ensemble that models a quantum chaotic system, denoted by $\langle\ldots\rangle_H$, the supersymmetry method extracts moments of the ensemble from ratios of determinants

$$D_n(X) = \left\langle \frac{\det(x_1+H)\det(x_2+H)\ldots\det(x_n+H)}{\det(\mathsf{x}_1+H)\det(\mathsf{x}_2+H)\ldots\det(\mathsf{x}_n+H)} \right\rangle_H. \tag{1}$$

The complex variables $x_i = -E_i \pm i\eta$ and $\mathsf{x}_i = -\mathsf{E}_i \pm i\eta$ contain real energy arguments as well as infinitesimal imaginary offsets which mark the causal structure of the correlator[1] [20,21]. By taking derivatives of $D_n(X)$ and setting the energy arguments equal, one obtains $n$th moments[2] of the spectral density $\rho(E) = \mathrm{tr}\,\delta(E-H)$. The method is called 'supersymmetric' because the determinants and inverse determinants can be represented by integrals over fermionic and bosonic vector degrees of freedom, manifesting an underlying U($n|n$) supergroup structure. In the limit where the energy differences $\Delta E = |E_i - E_j|$ become small and approach the mean

---

[1]We will always use serif variables for arguments of determinants, and sans serif for inverse determinants.
[2]For example, from $\mathcal{D}_2(X)$ one extracts the density-density correlator, which after Laplace transform to $\langle Z(\beta_1)Z(\beta_2)\rangle$ and analytic continuation of the inverse temperatures yields the spectral form factor.

level spacing $\Delta$, the global $U(n|n)$ is spontaneously broken to $U(\frac{n}{2}|\frac{n}{2}) \times U(\frac{n}{2}|\frac{n}{2})$ and the correlator Eq. (1) reduces to a non-linear $\sigma$-model [22] on the coset manifold $\text{AIII}_{n|n} \equiv \frac{U(n|n)}{U(\frac{n}{2}|\frac{n}{2}) \times U(\frac{n}{2}|\frac{n}{2})}$ [23]:

$$D_n(X) \simeq \int_{\text{AIII}_{n|n}} dQ \, \exp\left[i\frac{\pi}{\Delta}\text{str}(XQ)\right]. \tag{2}$$

This $\sigma$-model universally captures the late time physics of any quantum chaotic system: it only depends on the mean level spacing $\Delta$ and the symmetry class of the ensemble.[3] For JT gravity, the mean level spacing at energy $E$ is determined to first order in $e^{-S_0}$ by the Schwarzian density of states [25]

$$\Delta^{-1} = \frac{e^{S_0}}{4\pi} \sinh\sqrt{2\pi E}. \tag{3}$$

This is a perturbative input derived from semi-classical gravity. Crucially, we will show that JT gravity, *non-perturbatively* completed by its universe field theory, is capable of reproducing the full $\sigma$-model Eq. (2), thus demonstrating its quantum chaotic nature.

**From universe field theory to the $\sigma$-model of quantum chaos**   To derive the $\sigma$-model directly from JT universe field theory, we study correlation functions of vertex operators $e^{\Phi(x)}$ and $e^{-\Phi(x)}$, which can be seen as the double-scaled analogs of the determinant and inverse determinant operators in Eq. (1). The field $\Phi(x)$ is a $\mathbb{Z}_2$-twisted chiral boson living on the JT spectral curve, as will be reviewed in Section 2.2. Previously, we showed that correlation functions of the current $\partial\Phi$ compute all-genus multi-boundary wormhole amplitudes in JT gravity, after inverse Laplace transform [15]. This time we take an inverse Laplace (or Fourier) transform of the vertex operators $e^{\pm\Phi(x)}$ to demonstrate, in Section 3, that their correlation function can be rewritten as a *flavor matrix integral* over the space of Hermitian supermatrices in $\text{GL}(n|n)$:

$$\left\langle\left\{e^{\Phi(x_1)}e^{-\Phi(x_1)}\cdots e^{\Phi(x_n)}e^{-\Phi(x_n)}\right\}\right\rangle_{\text{KS}} = \int_{(n|n)} dA \, \exp\left[-e^{S_0}\Gamma(A) + e^{S_0}\text{str}(XA)\right]. \tag{4}$$

Here the curly brackets denote a normal ordering prescription for the product of vertex operators, and the angular brackets denote the expectation value in Kodaira-Spencer (KS) universe field theory. On the right-hand side, the matrix potential $\Gamma(A)$ can be computed perturbatively, to arbitrary order in $e^{-S_0}$, from the topological recursion relations satisfied by $\langle\partial\Phi(x_1)\ldots\partial\Phi(x_n)\rangle_{\text{KS}}$. To leading order, it is given by the single supertrace $\Gamma(A) = \text{str}\,\Gamma_0(A)$, where the functional form of $\Gamma_0(y)$ is determined by solving the spectral curve equation $H(x, y) = 0$ and integrating $-x\,dy$, the one-form dual to the canonical holomorphic one-form $\omega = y\,dx$, giving

$$\Gamma_0(y) = -\int^y x(y')dy'. \tag{5}$$

In the case of the 'Airy' spectral curve, $H(x, y) = y^2 - x$, which governs the low energy behavior of all topological gravity models [26], the potential is cubic $\Gamma(A) = \frac{1}{3}\text{str}(A^3)$, and the flavor matrix integral becomes a graded version of the celebrated Kontsevich matrix model [27]. For JT gravity, the spectral curve is derived from the Schwarzian density of states Eq. (3), and is given by $H(x, y) = y^2 - \frac{1}{(4\pi)^2}\sin^2(2\pi\sqrt{x})$. Like the Kontsevich model, one has to select appropriate integration contours for the eigenvalues of $A$, which are analyzed in Appendix A.

---

[3]In the above discussion we have assumed the ensemble to be unitary invariant, but there are in total 10 distinct symmetry classes, following the classification of [24].

After establishing the duality Eq. (4), we perform a stationary phase analysis of the flavor matrix integral in the limit that the probe energies approach each other, $\Delta E \to 0$, and the mean level spacing $\Delta$ (and hence $e^{-S_0}$) goes to zero, while keeping their ratio fixed to

$$s = \Delta E / \Delta. \tag{6}$$

Since $\Delta E \to 0$, we are probing the very late time behavior of the system. In this 'late time limit' there is a whole saddle-point manifold over which one should integrate, which turns out to be precisely the coset manifold $\mathrm{AIII}_{n|n}$. In fact, we show that the flavor matrix theory reduces to the non-linear $\sigma$-model of quantum chaos in the late time limit

$$\int_{(n|n)} dA \exp\left[-e^{S_0}\Gamma(A) + e^{S_0}\mathrm{str}(XA)\right] \xrightarrow[\text{phase}]{\text{stationary}} \int_{\mathrm{AIII}_{n|n}} dQ \exp\left[i\frac{\pi}{\Delta}\mathrm{str}(XQ)\right], \tag{7}$$

where $\Delta^{-1}$ is given by the disk JT density of states Eq. (3). This result demonstrates that the completion of JT gravity in terms of its universe field theory knows both about the perturbative (in $e^{-S_0}$) sum over topologies, as well as the fully non-perturbative late time ergodic physics. For example, when $s \gg 1$, the main contribution to the $\sigma$-model comes from a perturbative expansion around the standard saddle. For $s \gtrsim 1$, a new class of supersymmetry breaking saddles becomes important, the so-called Andreev-Altshuler saddle points [28], as described in [18]. When $s < 1$, one needs to integrate $Q$ over the full Goldstone manifold, corresponding to a phase where causal symmetry is restored. As one can see, each of these phases is captured by universe field theory, and so the result Eq. (7) improves on [15] where similar vertex operator calculus was used to derive the sine kernel in JT gravity.

**Late time quantum chaos: the open string perspective**   It may seem like a miracle that a theory of semi-classical gravity should be sensitive to the late time quantum chaotic properties of the underlying microscopics. Our aim is to give a gravitational interpretation of this result, using intuition from (topological) string theory.[4] The main idea is to access non-perturbative information (in $e^{-S_0}$) by allowing JT strings to end on D-branes embedded in a higher dimensional target space $\mathsf{CY}$. This six-dimensional Calabi-Yau is a fibration of the spectral curve, defined by

$$uv - H(x, y) = 0, \tag{8}$$

where $u, v, x, y \in \mathbb{C}$. To the Calabi-Yau one associates a holomorphic $(3,0)$-form

$$\Omega = \frac{du}{u} \wedge dx \wedge dy. \tag{9}$$

We can wrap a one-complex-dimensional brane on the non-compact submanifold $u = 0$, which is parametrized by $v$ and a point on the spectral curve. Similarly, we define anti-branes by wrapping them on $v = 0$. From $uv = 0$ we see that $\frac{du}{u} = -\frac{dv}{v}$, so that exchanging $u$ and $v$ amounts to a change of sign for $\Omega$. This shows that anti-branes can be viewed as branes with the opposite orientation. Inserting a vertex operator $e^{\Phi(x)}$ creates a brane, while $e^{-\Phi(x)}$ creates an anti-brane, above a point $x$ on the spectral curve. The real part of $x$ is interpreted as an energy $E$ in the Schwarzian theory, which is kept fixed by imposing fixed energy boundary conditions on the JT gravity action [35,36]. On the level of JT gravity, the fixed energy boundary term is the Legendre transform of the usual Dirichlet boundary term for fixed inverse temperature $\beta$. So, in the universe field theory language, inserting a vertex operator $e^{\Phi(x)}$ creates a brane in target space on which JT worldsheets with boundary energy $E$ can end.

---

[4]These ideas trace back to the seminal work of Dijkgraaf and Vafa on topological strings and matrix models [29–32], as well as work by Maldacena, Moore, Seiberg and Shih on minimal string theory [33,34].

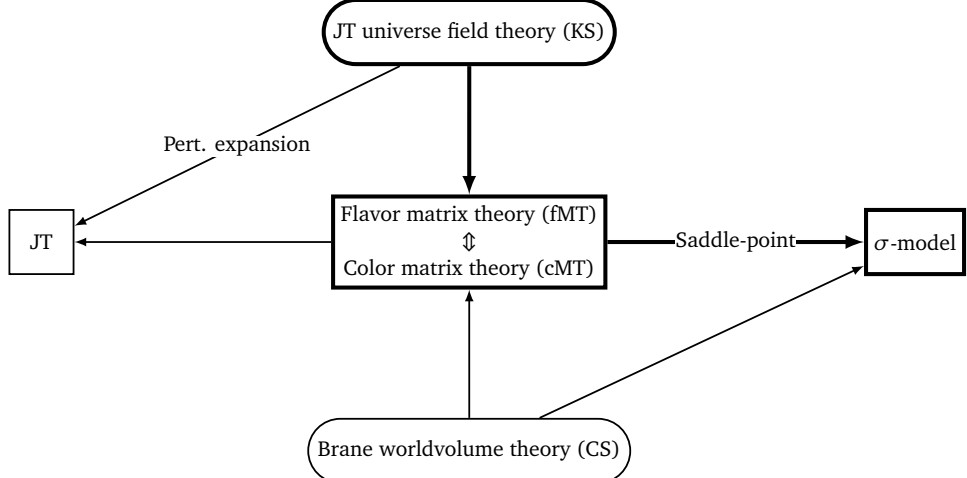

Figure 1: Diagram of the related theories discussed in this article. (Left) JT gravity, defined perturbatively as a sum over topologies weighted by $(e^{-S_0})^{2g-2+n}$. It may be completed non-perturbatively by a large $L$ color matrix theory (middle), double-scaled to the spectral edge. This is dual to a flavor matrix integral, which can be derived exactly from JT universe field theory (top) or from the brane worldvolume theory (bottom). A saddle point approximation of fMT then leads to the $\sigma$-model of quantum chaos.

This D-brane point of view gives a simple explanation for the appearance of the flavor matrix theory Eq. (4). Namely, inserting $n$ pairs of vertex operators $e^{\Phi(x_i)}e^{-\Phi(\mathsf{x}_i)}$ creates $n$ brane/anti-brane pairs. When we consider the limit of coincident probe energies, $x_i \to \mathsf{x}_i$, we get a stack of branes and anti-branes and the gauge group enhances to U$(n|n)$ [37]. This is precisely the 'causal symmetry' alluded to in the context of quantum chaos. Indeed, we show in Section 4.1 that the effective brane worldvolume theory is to leading order in $e^{-S_0}$ equal to the flavor matrix integral. The small imaginary offsets of the brane positions $x_i$ spontaneously break the U$(n|n)$ causal symmetry and finite energy differences $\Delta E$ give rise to massive modes – this is the well-known Higgs effect for D-branes.

The open string perspective also provides an explanation for the color-flavor duality which connects the double-scaled matrix model of Saad, Shenker and Stanford to the Kontsevich-like flavor matrix integral presented in this article. This duality has been established using the supersymmetry method and a generalized Hubbard-Stratonovich transformation in [38, 39]. In Section 4, we give this duality an interpretation using the open string field theory in the target space CY. Namely, the degrees of freedom of the flavor matrix theory are open JT strings ending on the non-compact branes introduces above, whereas those of the color matrix theory are open JT strings ending on *compact* branes, described in detail in Section 4.2, which wrap the blown-up singularities of the spectral curve.

**Connection to SYK** Finally, our results establish an interesting new connection to the SYK model. As is well known, the SYK model [40, 41] reduces at low energies to the Schwarzian theory [42], which is the same as JT gravity on the disk. This relates JT and SYK on the most coarse-grained level, or at early times. Interestingly, we have found that also the very late time description of (non-perturbatively completed) JT quantum gravity, namely the non-linear $\sigma$-model Eq. (14), is precisely the same as the late time ergodic phase of the SYK model derived in [43]. Of course, this does not prove a full duality between JT and SYK, but it shows that they are in the same universality class: they have the same mean level spacing Eq. (3) and the same pattern of causal symmetry breaking.

**Outline**

This article is meant to bridge a gap between the communities of quantum chaos and holography. We have therefore summarized some necessary background in Section 2: first, flavor matrix theory is introduced as an organizing principle for quantum chaos, followed by a short primer on JT universe field theory. A more comprehensive treatment of both topics can be found in our previous articles [15, 18]. After setting the scene, we derive the non-linear $\sigma$-model of quantum chaos directly from JT universe field theory in Section 3. This is accomplished by inserting brane/anti-brane operators and Fourier transforming them to a flavor matrix integral, whose saddle-point approximation in the late time limit is the sought-after $\sigma$-model. In Section 4, we interpret this result by studying the brane worldvolume theory, whose effective description when the branes coincide is given by a dimensional reduction of U($n|n$) holomorphic Chern-Simons theory. By identifying both compact and non-compact branes in the target space geometry, we give an open string interpretation of the color-flavor map. In Appendix A we perform a stationary phase analysis of the flavor matrix integral and select the defining integration contours. It is shown how Stokes' phenomena lead to causal symmetry breaking, which allows us to identify which saddle points contribute. A schematic overview of the relevant concepts and their interrelations is presented in Figure 1.

## 2 Setting the scene

### 2.1 The nonlinear $\sigma$-model of quantum chaos

One of the beautiful aspects of ergodic chaotic quantum systems is that they essentially all behave the same way. This phenomenon is often paraphrased as *random matrix universality*: "Quantum systems that are classically chaotic are equivalently described by random matrix theory at large time scales" [44]. Here, 'large time scales' refers to scales longer than the time it takes to establish ergodic equilibration of the dynamics, called the Thouless time $t_{\mathrm{T}}$. This scale is non-universal, and needs to be determined on a system-to-system basis. The longest characteristic time scale is the Heisenberg time (aka *plateau time*), $t_{\mathrm{H}} = \Delta^{-1}$, where $\Delta = \langle \rho(E) \rangle^{-1}$ is the average microstate spacing at the chosen probe energy $E$. For generic chaotic quantum systems, $\langle \rho(E) \rangle$ of a system with a total number of $L$ microstates exhibits smooth dependence on $E$, see Fig. 2.

One might object that the statement of random matrix universality is somewhat of a tautology. Conceptually, a random matrix Hamiltonian is just another representative in the class of chaotic quantum systems. So the statement is but repeating that they all behave identically. A more substantial characterization is as follows: chaotic quantum systems spontaneously break a continuous symmetry related to the causal structure of time evolution. In ergodic quantum dynamics, this symmetry gets restored after Heisenberg time $t_H$. This symmetry breaking principle finds its quantitative formulation in a simple mean field theory, which takes the form of a non-linear $\sigma$-model. Much as mean field theory for, say, a magnetization order parameter captures the universal features of of ferromagnetism, the $\sigma$-model describes the universality class of ergodic quantum chaos.

In order to explain the symmetry breaking principle in general terms, let us write the determinant ratio from the Introduction, Eq. (1), as a superdeterminant

$$D_n(X) = \langle \mathrm{Sdet}(X \otimes \mathbb{1}_{\mathrm{c}} + \mathbb{1}_{\mathrm{f}} \otimes H) \rangle_H \equiv \langle \mathrm{Sdet}(\Xi) \rangle_H \,. \tag{10}$$

We think of $\Xi$ as an operator acting in a product Hilbert space $\mathcal{H} = \mathcal{H}_{\mathrm{f}} \otimes \mathcal{H}_{\mathrm{c}}$ of a $2n$-dimensional graded 'flavor-space' $\mathcal{H}_{\mathrm{f}} = \mathbb{C}^{n|n}$ and $L$-dimensional 'color space' $\mathcal{H}_{\mathrm{c}} = \mathbb{C}^L$. As such, $\Xi$ carries an adjoint representation under the group U($nL|nL$). Transformations under this group change

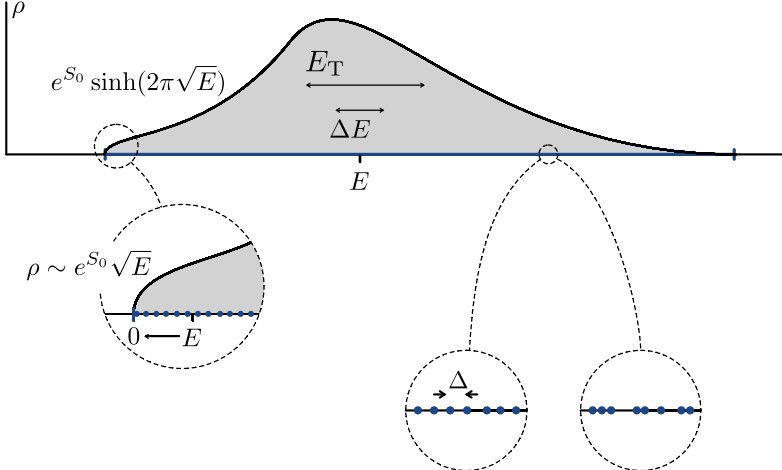

Figure 2: Spectral density of a chaotic quantum system. A total number of $L$ microstates is contained in a compact spectral support defined by a non-vanishing average spectral density $\rho(E)$. In the gravitational context, one is often interested in energies 'double scaled' to the ground state, $E = 0$, inset left. Different from the energy levels of a generic system (right inset), levels of chaotic systems are almost uniformly spaced (middle) and cannot 'touch'.

$\Xi \rightarrow U \Xi U^{-1}$ but naturally leave our determinant correlation functions invariant. This huge symmetry group possesses two interesting subgroups, the color group $U_c = \mathbb{1}_f \otimes U(L)$ which acts on $H \rightarrow U H U^{-1}$, leaving $X$ invariant, and the $2n$ dimensional flavor group $U_f = U(n|n) \otimes \mathbb{1}_c$, changing $X \rightarrow T X T^{-1}$, but leaving $H$ invariant. The interplay of the dual pair defined by the color and the flavor symmetry group is key to the characterization of universality in quantum chaos.

The idea behind the construction of effective field theories of quantum chaos is to turn the complexity of the theory in color space into an advantage: Averaging over realizations of $H$ will eradicate contributions to the spectral determinant that fluctuate strongly under the action of the color group.[5] Eventually, in ergodic limits, only contributions in the color singlet representation survive. In this projection onto the color singlet sector, the measure associated to the integration over the $H$-measure gets converted into an integration over flavor degrees of freedom. This *color-flavor* duality assumes its purest form in the case of invariant matrix ensembles, where $\langle \ldots \rangle_H = \int dH \exp(-L \operatorname{tr} V(H))$, with a potential function $V(H)$. It can be shown that [18]

$$\langle \operatorname{Sdet}(X \otimes \mathbb{1}_c + \mathbb{1}_f \otimes H) \rangle_H = \langle \operatorname{Sdet}(X \otimes \mathbb{1}_c + A \otimes \mathbb{1}_c) \rangle_A \,, \tag{11}$$

where $A \in \operatorname{GL}(n|n)$ is a flavor matrix, with $n$ 'bosonic' and $n$ 'fermionic' eigenvalues, and the flavor matrix integral is $\langle \ldots \rangle_A = \int dA \exp(-L \operatorname{str} W(A))$. In the Gaussian case, $V(H) = H^2$, the flavor matrix potential $W(A) = A^2$ is also quadratic in $A$ [45]. For general potentials, the color and flavor ensembles agree on the level of the generating function [18]. The representation on the right then defines the starting point for the construction of an effective flavor matrix

---

[5]The minimal averaging measures required to block color space fluctuations must be determined on a case to case basis. However, a rule of thumb is that for systems with ergodic chaotic dynamics, averages over parameter windows corresponding to just a few microstate spacings can already be sufficient.

theory. Besides $A$ being a low dimensional matrix, the advantage of this representation is that $\mathrm{Sdet}(X \otimes \mathbb{1}_\mathrm{c} + A \otimes \mathbb{1}_\mathrm{c}) = \mathrm{sdet}(X+A)^L = \exp(-L\,\mathrm{str}\,\ln(X+A))$, due to color isotropy: the flavor theory is amenable to a stationary phase analysis stabilized by the large parameter $L$.

To anticipate what is awaiting us in this large-$L$ analysis, let us go one step back to the theory before having taken any averages. For small differences between the probe arguments, $(x_i, \mathsf{x}_i)$, flavor symmetry is an approximate symmetry not just of the correlation functions but of the operator $\Xi$ itself: $T\Xi T^{-1} \approx \Xi$. In view of what has been said above, we expect these transformations to become soft degrees of freedom of the putative flavor matrix theory (fMT). To understand the structure of these theories, it is key to realize that they all (irrespective of the detailed realization of $H$) live under the spell of a symmetry breaking principle. In physically meaningful correlation functions, flavor symmetry is never realized in a strict sense: we need our probe arguments infinitesimally shifted into the complex plane as, say, $\mathrm{Im}(x_j) = \pm i\eta$, and the same with the probes in the denominator, $\mathsf{x}_i$. Even in the limit $\mathrm{Re}(x_j) = \mathrm{Re}(\mathsf{x}_j) = -E$, flavor symmetry remains infinitesimally broken by these increments. Referring to their physical meaning as indicators of causality, we refer to the approximate symmetry under flavor transformations as *causal symmetry* of the theory.

This is as much as can be said in the most general terms; no reference to chaos or concrete realizations of $H$ is made so far. However, let us now turn back to the role played by the ensemble average $\langle \ldots \rangle_H$ over microscopically different realizations of any model. The eigenvalues of individual $H$'s define a set of finely spaced, yet isolated singularities of $\Xi$ in the complex $x-$plane. Averaging over an ensemble will blur these point singularities into a cut structure, see Fig. 2. On the level of a crudest 'mean field' approximation applied to the averaged theory, we expect the correlation function to assume the form

$$\langle \mathrm{Sdet}(\Xi) \rangle_H \longrightarrow \mathrm{Sdet}\left( (X + i\gamma\tau_3) \otimes \mathbb{1}_\mathrm{c} + \mathbb{1}_\mathrm{f} \otimes H_0 \right), \tag{12}$$

where $H_0$ is a possible un-averaged contribution to $H$, $\gamma = \gamma(E)$ a finite imaginary offset non-vanishing along the support set of the theory (the interval(s) of $E$ for which $\rho(E) > 0$), and $\hat{\tau}_3 = \tau_3 \otimes \mathbb{1}_\mathrm{f}$ is a flavor Pauli matrix with sign structure set by the $\eta$-parameters.[6] The key point here is that the infinitesimal $\eta$ sets the sign of the finite $\gamma$: causal symmetry is spontaneously broken at the mean field level inside the spectral curve. Still, no reference to chaos is made: averaging over realizations of an integrable theory defines a cut structure too.

It is natural to expect that in a theory with large symmetry group, the mean field configuration will be subject to fluctuations. To get an idea of their influence, consider the full symmetry group action of $U(nL|nL)$ on Eq. (12). Such transformations preserve eigenvalues, and hence are compatible with the analytic structure of the theory. However, at this point, differences between integrable and chaotic parent theories begin to show. In the former case, fluctuations with non-trivial color space structure may be physically significant even at large time scales (provided they commute with $H_0$.) However, in a chaotic ergodic phase the above mentioned projection onto the color singlet sector becomes effective: Only $T \in \mathrm{U}_\mathrm{f}$ remains in the fluctuation spectrum, i.e. we are reduced to the degrees of freedom of fMT. In fact, a stronger statement can be made. For $n$ even, the effective degrees of freedom assume the form

$$Q \equiv T\hat{\tau}_3 T^{-1} \in \frac{\mathrm{U}(n|n)}{\mathrm{U}(\frac{n}{2}|\frac{n}{2}) \times \mathrm{U}(\frac{n}{2}|\frac{n}{2})}, \tag{13}$$

where the divisor represents the unbroken symmetry group (transformations commuting with $\hat{\tau}_3$). As in the introduction, we denote the coset supersymmetric target space with Cartan's notation $\mathrm{AIII}_{n|n}$. Referring for a more detailed discussion to Section 3.3, these coset degrees of freedom arise as the reduction of fMT to a non-linear $\sigma$-model. For finite differences between

---

[6]Without loss of generality, we assume as many positive as negative increments $i\eta$.

the probe arguments contained in $X$, the fluctuations $Q$ acquire a mass. To lowest order in this explicit symmetry breaking, we will end up with a fluctuation integral

$$D_n(X) \simeq \int_{\mathrm{AIII}_{n|n}} dQ\, e^{iS[Q]}, \qquad S[Q] \equiv \frac{\pi}{\Delta}\mathrm{str}(XQ), \tag{14}$$

where $\Delta = \langle\rho(E)\rangle^{-1}$ is the averaged spectral density at the center value $E$ (the only characteristic energy scale in the problem). This is the non-linear $\sigma$-model mentioned in the introduction. It provides a complete description of the ergodic phase of quantum chaos.

To see how, let us discuss the role of the flavor coset space fluctuations described by Eq. (14). In the limit of small energy differences $|E_i - E_j| \equiv \Delta E \sim \Delta$, large fluctuations signal that the 'true configurations' of the theory are far detached from the naive cut-saddle points. These fluctuations act to restore the previously broken causal symmetry. Remembering that causal symmetry breaking was equivalent to the emergence of a cut structure along the spectral curve, the restoration of this symmetry in the limit of small energy difference, or large times, must amount to the re-emergence of information on the discrete pole structure of the chaotic spectrum. Indeed, one can do the $Q$ integral in closed form to verify that it produces the exact correlation functions ('ramp+plateau') of ergodic quantum chaos.

En route to the deep limit $\Delta E \lesssim \Delta$, one encounters various physically interesting intermediate structures: for $\Delta E \gg \Delta$ the perturbative expansion around the 'standard saddle point' $Q_{st} = \hat{\tau}_3$ defines an asymptotic series equal to the perturbative expansion of other effective theories of quantum chaos. Specifically, it can be shown to be identical to the topological expansion of conventional color matrix theory (more precisely to the limit of that expansion for small differences $\gamma \gg \Delta E \gtrsim \Delta$), or to the mini-universe expansion of JT gravity in the same limit. For $\Delta E \sim \Delta$, and $n = 2$, a second, supersymmetry breaking saddle point $Q_{AA} = \tau_3 \otimes \tau_3$ begins to play a rôle. This saddle point is known as the Altshuler-Andreev saddle, and related to a standard saddle by a discrete (Weyl group) transformation in $U_f$ [18].

Summarizing, a combination of phenomenological arguments and symmetry considerations identifies the $\sigma$-model Eq. (14) as the effective theory of the quantum ergodic phase. An implicit assumption in this construction was that our parent theory, $H$, contains no anti-linear symmetries besides hermiticity. More generally, one needs to distinguish between ten different classes of anti-linear symmetries, and in the consequence ten different incarnations of fMT's [24]. All have in common that they assume the form of integrals over low dimensional 'classical' supergroups or -coset spaces. In view of the generality of the construction, one expects *any* theory describing an ergodic quantum phase must collapse to one of these variants in the long time limit. In this paper, we demonstrate this reduction principle for a family of theories of two-dimensional gravity that includes JT gravity. To understand how this happens, we first introduce universe field theory, which will be our main tool connecting JT to quantum chaos.

## 2.2 The universe field theory of JT gravity

The main idea underlying the KS/JT identification is to view the JT universes as string worldsheets propagating in the target space Calabi-Yau manifold [14]

$$\mathrm{CY}: \quad uv - y^2 + \frac{1}{(4\pi)^2}\sin^2(2\pi\sqrt{x}) = 0. \tag{15}$$

The closed string field theory associated to the topological strings propagating in the geometry Eq. (15) is the six-dimensional KS theory [46] of complex structure deformations. Given this data, one can apply a dimensional reduction of Eq. (15) to the *spectral curve* $\mathcal{S}_{\mathrm{JT}}$ defined by

$$\mathcal{S}_{\mathrm{JT}}: \quad H(x, y) \equiv y^2 - \frac{1}{(4\pi)^2}\sin^2(2\pi\sqrt{x}) = 0. \tag{16}$$

We think about the spectral curve (and consequently the target space of the topological string) as arising from the continuous Schwarzian density of states [25]

$$\rho_0(E) = \frac{1}{4\pi} \sinh(2\pi\sqrt{E}),$$
(17)

by using the identification $x = -E$. This function determines 'disk' density of states in JT as $\langle \rho(E) \rangle_{\mathsf{KS}}^{(g=0)} = e^{S_0} \rho_0(E)$ [3, 15]. More generally, a different choice of spectral curve $\mathcal{S}$ leads to a duality between KS theory and other models of two-dimensional gravity, like topological gravity (the choice $y \sim \sqrt{x}$) or minimal string theory ($y \sim x^{p/2} + \ldots$).

The fundamental KS field is a $\mathbb{Z}_2$-twisted boson $\Phi$ on $\mathcal{S}_{\mathrm{JT}}$, with a branch cut at the negative real axis $x \in (-\infty, 0]$. It describes the complex structure deformations of $\mathcal{S}_{\mathrm{JT}}$ through

$$\delta\omega = d\Phi,$$
(18)

where $\omega = y\, dx$ is a holomorphic $(1,0)$-form. We often work in the uniformizing coordinate $z$, which parametrizes the spectral curve as $x(z) = z^2$, $y(z) = \frac{1}{4\pi} \sin(2\pi z)$, so that the field $\Phi(z)$ is single valued and odd in $z$. There is another field $\mathcal{J}$, which can be identified with $\mathcal{J} = d\Phi$ on-shell, and which appears in the cubic interaction vertex of the theory

$$S_{\mathrm{int}} = \frac{\lambda}{2} \oint_0 \frac{\Phi}{\omega} \mathcal{J}^2.$$
(19)

This interaction is a boundary term, localized on a small contour encircling the branch point at $z = 0$. As explained in [15], it can be viewed as arising from a $\Phi$-dependent coordinate transformation relating the complex structure at the origin and infinity. The 'closed string' coupling constant $\lambda$ is interpreted in JT gravity as the genus expansion parameter

$$\lambda = e^{-S_0}.$$
(20)

From the point of view of gravity, the KS theory is a 'universe field theory' or 'wormhole field theory', in the sense that its correlation functions compute gravitational wormhole contributions in JT. One can check this explicitly by using the Schwinger-Dyson equations of the KS theory (derived in [15]). As an example of how the dictionary works, we compute the JT path integral on a three-holed sphere from the tree level three-point function of KS. In terms of the uniformizing coordinate $x(z) = z^2$ the connected three-point function at tree level is computed as

$$\langle \mathcal{J}(z_0)\mathcal{J}(z_1)\mathcal{J}(z_2) \rangle_{\mathsf{KS}}^{(g=0),c} = \frac{\lambda}{2} \oint_0 \frac{dz}{2\pi i} \frac{\mathsf{G}(z_0,z)}{\omega(z)} \left\langle \overbrace{\mathcal{J}(z_1)\mathcal{J}(z_2)\{\mathcal{J}(z)\mathcal{J}(z)\}} \right\rangle$$
(21)

$$= \lambda \oint_0 \frac{dz}{2\pi i} \frac{\mathsf{G}(z_0,z)}{\omega(z)} \mathsf{B}(z_1,z)\mathsf{B}(z_2,z),$$
(22)

where at the first equality we have used the Schwinger-Dyson equation to replace $\mathcal{J}(z_0)$ by

$$\frac{\lambda}{2} \oint_0 \frac{dz}{2\pi i} \frac{\mathsf{G}(z_0,z)}{\omega(z)} \{\mathcal{J}(z)\mathcal{J}(z)\},$$
(23)

and $\{\cdots\}$ denotes the 'normal ordering' operation of subtracting the OPE divergence. If we now plug in the form of the one-point function $\omega(z)$ and the (free) propagators [15]

$$\mathsf{G}(z_0,z) = \langle \Phi(z_0)\mathcal{J}(z) \rangle = \frac{1}{z_0 - z} - \frac{1}{z_0 + z},$$
(24)

$$\mathsf{B}(z_i,z) = \langle \mathcal{J}(z_i)\mathcal{J}(z) \rangle = \frac{1}{(z_i - z)^2} + \frac{1}{(z_i + z)^2},$$
(25)

we find that the genus 0 connected three-point function $\langle \mathcal{J}(z_0)\mathcal{J}(z_1)\mathcal{J}(z_2)\rangle_{\mathsf{KS}}^{(g=0),\mathsf{c}}$ is given by

$$\mathrm{Res}_{z=0}\frac{\lambda}{z_0^2 - z^2}\frac{\pi}{\sin(2\pi z)}\mathsf{B}(z_1,z)\mathsf{B}(z_2,z) = \frac{\lambda}{z_0^2 z_1^2 z_2^2}. \tag{26}$$

The dictionary to JT gravity consists of defining boundary creation operators $Z(\beta)$, which are obtained from the KS field $\mathcal{J}(z)$ by inverse Laplace transform (in $z^2$),

$$Z(\beta) = \int_{c-i\infty}^{c+i\infty}\frac{dz}{2\pi i}e^{\beta z^2}\mathcal{J}(z). \tag{27}$$

We can compute their free three-point function using Eq. (26) and doing the inverse Laplace transform we obtain

$$\langle Z(\beta_1)Z(\beta_2)Z(\beta_3)\rangle_{\mathsf{KS}}^{(g=0),\mathsf{c}} = \frac{\lambda}{\pi^{3/2}}\sqrt{\beta_1\beta_2\beta_3}. \tag{28}$$

This is precisely the answer for the wormhole contribution in JT gravity. Namely, Ref. [3] showed that the JT gravitational path integral on a spacetime wormhole with fixed topology is

$$\mathcal{Z}_{g,n}(\beta_1,\dots,\beta_n) = \left(e^{S_0}\right)^\chi \int_0^\infty \prod_{i=1}^n d\ell_i\,\ell_i Z_{\mathrm{trumpet}}(\beta_i,\ell_i)V_{g,n}(\ell_1,\dots,\ell_n), \tag{29}$$

where $\chi = 2 - 2g - n < 0$ is the Euler characteristic of the wormhole, and the boundary graviton mode is represented by a trumpet partition function

$$Z_{\mathrm{trumpet}}(\beta,\ell) = \frac{1}{\sqrt{4\pi\beta}}e^{-\ell^2/(4\beta)}. \tag{30}$$

The volume associated to the non-trivial bulk metrics is given by the Weil-Petersson volumes $V_{g,n}(\ell_1,\dots,\ell_n)$ associated to a surface of constant negative curvature. For the example at hand, we need the Weil-Petersson volume of the 'pair-of-pants', which is $V_{0,3} = 1$. Gluing the three trumpets, we therefore find:

$$\mathcal{Z}_{0,3}(\beta_1,\beta_2,\beta_3) = e^{-S_0}\int_0^\infty \prod_{i=1}^3 d\ell_i\,\frac{\ell_i}{\sqrt{4\pi\beta_i}}e^{-\ell_i^2/(4\beta_i)} = \frac{e^{-S_0}}{\pi^{3/2}}\sqrt{\beta_1\beta_2\beta_3}. \tag{31}$$

This agrees with the result in Eq. (28). This is an example of the more general dictionary between Euclidean wormholes in JT and $\mathcal{J}$-insertions in the universe field theory, mediated by the inverse Laplace transform:

$$\mathcal{Z}_{g,n}(\beta_1,\dots,\beta_n) = \lambda^\chi \int_{c-i\infty}^{c+i\infty}\prod_{i=1}^n dz_i\,e^{\beta_i z_i^2}\,\langle\mathcal{J}(z_1)\cdots\mathcal{J}(z_n)\rangle_{\mathsf{KS}}^{(g),\mathsf{c}}. \tag{32}$$

The above identification follows from a careful study of the recursion relations that are satisfied on both sides. Using the explicit expression Eq. (29), one can show that the wormholes with different Euler characteristic are related to each other through a version of Mirzakhani's recursion relation [47,48] for the Weil-Petersson volumes. On the other hand, the KS correlation functions satisfy a Schwinger-Dyson equation that is equivalent to the topological recursion relations [26,49,50] of Eynard and Orantin (as was first observed in [32]). Using the fact that Mirzhakani's recursion and the topological recursion with initial data the JT spectral curve Eq. (16) agree after a Laplace transform, cf. [51], the relation in Eq. (32) follows. Much more can be said about KS theory, but we choose to introduce the relevant features as we go along in the derivation of the fMT in the next section.

# 3 fMT from universe field theory

In this section we derive a flavor matrix theory from brane creation operator insertions in KS universe field theory. Our construction of the flavor matrix theory in this section proceeds in three steps. In Section 3.1 we introduce vertex operators in KS theory. Seen through the lens of the flavor matrix model, they probe eigenvalue correlations along the spectral curve. From the target space point of view, they create branes and anti-branes. Either way, they play the same the role as the determinant operators in Eq. (1), but instead of averaging over large color matrices, we are computing a Euclidean correlation function in KS field theory. We show in Section 3.2 that the correlator of brane/anti-brane vertex operators leads to an eigenvalue representation of a flavor matrix integral. The crucial ingredient is to use the transformation properties of $e^{\pm\Phi(x)}$ under symplectic transformations. Having identified the fMT of JT gravity, the stationary phase analysis of this integral (Section 3.3) naturally gives rise to the nonlinear $\sigma$-model discussed in Section 2.1.

## 3.1 (Anti-)brane creation operators

In Section 2.2 we have argued that semiclassical JT gravity is captured by the perturbative expansion of the KS field theory on the spectral curve $\mathcal{S}_{JT}$. However, our goal is to show that the fully *non-perturbative* physics of the theory is described by a fMT, and upon further reduction the universal non-linear $\sigma$-model Eq. (14) presented in Section 2.1.

As a first step towards realizing the fMT theory in the KS framework, we introduce D-brane-like objects in the target space geometry assuming the role of the the probe determinants in Eq. (1). In fact, the B-model topological string theory on Eq. (15) allows for certain non-compact branes that do precisely that (see [31, 52], where they are referred to as 'B-branes'): they probe a particular 'eigenvalue' in the spectral $x-$plane. Topological (anti-)branes wrapped around the submanifold

$$\mathcal{B}: u = 0, \qquad (x, y) \in \mathcal{S}_{JT}, \tag{33}$$

in the Calabi-Yau Eq. (15) give rise to vertex operators[7]

$$\psi(x) = e^{\Phi(x)}, \qquad \psi^\dagger(x) = e^{-\Phi(x)}, \tag{34}$$

on the spectral curve. Given the identification $\text{Re}(x) = -E$, we think about the insertion of a fermionic field in Eq. (34) as defining a topological (anti-)brane, on which JT universes with 'fixed energy boundaries' can end. The precise boundary condition for the JT 'worldsheet' theory is Dirichlet-Neumann (DN), where one fixes the dilaton and its normal derivative at the boundary. (Various choices of boundary conditions in JT gravity, including DN, are nicely summarized in [36].) This is to be contrasted with the Dirichlet-Dirichlet (DD) boundary condition, that leads to the canonical partition function Eq. (29) where one fixes the dilaton as well as boundary lengths $\beta_1, \beta_2, \ldots$. These correspond to temperatures in the Schwarzian boundary theory. On the level of the action, one can move between both choices of boundary condition by a suitable Legendre transform, so we can think about the DN boundary conditions as fixing the energies $E_1, E_2, \ldots$ in the boundary theory: it defines a microcanonical partition function. The path integral associated to the fixed energy boundaries is related to the fixed temperature boundaries by yet another inverse Laplace transform

$$\mathcal{Z}(E_1, \ldots, E_n) = \int_{c-i\infty}^{c+i\infty} \prod_{i=1}^{n} \frac{d\beta_i}{\beta_i} e^{\beta_i E_i} \mathcal{Z}(\beta_1, \ldots, \beta_n). \tag{35}$$

---

[7]These are normal ordered exponentials, meaning that the OPE divergences from $\Phi\Phi$ contractions have been subtracted. Technically, $\Phi$ should be understood here as the indefinite integral of $\mathcal{J}$.

For example, we can compute the DN partition function of the disk to be

$$Z_{\text{disk}}(E) = \int_{c-i\infty}^{c+i\infty} \frac{d\beta}{\beta} e^{\beta E} Z_{\text{disk}}(\beta) = e^{S_0} \int^E dE' \rho_0(E'),$$
(36)

which corresponds to the insertion of an integrated density of states. The DN boundaries conditions relate to the presence of so-called *energy-eigenbranes* (as defined in Ref. [35]) which fix a particular eigenvalue in the dual (color) matrix model.

The vertex operators in KS theory satisfy some useful properties. Firstly, they obey the boson-fermion correspondence [52], which states that

$$\lim_{z'\to z} \left\{ \psi(z)\psi^\dagger(z') \right\} = \partial\Phi(z).$$
(37)

Here the accolades signify subtracting the OPE singularity $\sim (z-z')^{-1}$. This property is the analog of the random matrix identity in which the derivative of a ratio of determinants gives the resolvent, upon taking the energy arguments equal. The role of the derivative, in the case at hand, is played by Wick's theorem in combining $\psi$ and $\psi^\dagger$ into a single normal ordered exponential. In [15], these brane/anti-brane insertions (on the two-sheeted spectral curve $\mathcal{S}_{\text{JT}}$) were used to study non-perturbative corrections to resolvent and spectral density correlation functions.

Secondly, the vertex operators transform under coordinate transformations of $x$ and $y$ that leave the symplectic form $dx \wedge dy$ invariant. Their transformation properties are inherited from the higher-dimensional closed string field theory. In short, the full six-dimensional KS theory on the Calabi-Yau Eq. (15) has a large symmetry group, namely diffeomorphisms that leave the holomorphic $(3,0)$-form invariant. Upon reduction to the spectral curve, the symmetry is broken to diffeomorphisms that preserve the symplectic form $dx \wedge dy$. The chiral boson $\Phi$ can be seen as the Goldstone boson for this broken symmetry. The broken symmetry generates Ward identities in the quantum theory, which coincide with the Schwinger-Dyson equations discussed above, cf. [31]. Diffeomorphisms that leave the symplectic form invariant are called *symplectomorphisms*, and the symplectic group of $\mathbb{C}^2$ is just $Sp(2,\mathbb{C}) \cong SL(2,\mathbb{C})$. So a symplectic transformation acts simply as

$$\begin{pmatrix} x' \\ y' \end{pmatrix} = \begin{pmatrix} a & b \\ c & d \end{pmatrix} \begin{pmatrix} x \\ y \end{pmatrix},$$
(38)

with $ad - bc = 1$. This leaves invariant $dx' \wedge dy' = dx \wedge dy$, whereas the holomorphic $(1,0)$-form $\omega = y dx$ changes up to a total derivative

$$y' dx' - y dx = dS.$$
(39)

The vertex operator $\psi(x) = e^{\Phi(x)}$ transforms under the symplectomorphism with a weight determined by the function $S(x,x')$ as

$$\widehat{\psi}(x') = \int \frac{dx}{\sqrt{\lambda}} e^{S(x,x')/\lambda} \psi(x).$$
(40)

Similarly, the anti-brane operator $\psi^\dagger(x)$ transforms with the opposite sign of $S(x,x')$. Here $\lambda = e^{-S_0}$ is the KS coupling constant. Eq. (40) can be interpreted by saying that the open string partition function transforms like a wave function, cf. [53–56]. We will be most interested in the 'S-transform', sometimes called 'x-y symmetry', which exchanges the coordinates $x' = y$ and $y' = -x$. In Ref. [57, 58] it is explicitly shown, using topological recursion, that this transformation is a symmetry of the KS partition function, to all orders in the genus expansion.

The observables $\psi, \psi^\dagger$ transform in a natural way under the $S$-transform. Namely, the classical action $S(x, x')$ corresponding to this transformation can be found using the definition Eq. (39), $x = -\partial_y S$ and $y = -\partial_x S$. This is solved by $S = -yx$, and so $\psi(x)$ and $\psi^\dagger(x)$ transform by Fourier (or inverse Laplace) transforms under the symplectic S-transformation

$$\widehat{\psi}(y) = \int \frac{dx}{\sqrt{\lambda}} e^{-xy/\lambda} \psi(x), \qquad \widehat{\psi^\dagger}(y) = \int \frac{dx}{\sqrt{\lambda}} e^{xy/\lambda} \psi^\dagger(x). \qquad (41)$$

Here the integration contours should be chosen such that the integrals converge — we will come back to this point momentarily. By inverting the above Fourier transform, we can similarly express $\psi(x)$ and $\psi^\dagger(x)$ in terms of the Fourier transformed (anti-)brane operators.

We can study the brane operators Eq. (41) by inserting them in KS correlation functions. Their perturbative expansion admits an 'open string' expansion

$$\langle \widehat{\psi}(y) \rangle_{\mathsf{KS}} = \exp\left[ -\frac{1}{\lambda} \sum_{n=0}^{\infty} \lambda^n \Gamma_n(y) \right]. \qquad (42)$$

To leading order in $\lambda$, the disk contribution to the 1-point function is simply $\exp \frac{1}{\lambda} \langle \widehat{\Phi}(y) \rangle_0$, where

$$\langle \widehat{\Phi}(y) \rangle_0 = -\int^y x(y') dy' \qquad (43)$$

is the integral of the dual one-form $-x \, dy$, and $x(y)$ is determined by the spectral curve equation $H(x, y) = 0$. Therefore, in the limit $\lambda \to 0$ we see that the Fourier transforms in Eq. (41) implement a Legendre transform of the disk potential

$$\langle \Phi(x) \rangle_0 = \int^x y(x') dx'. \qquad (44)$$

In the case that the spectral curve is given by $y^2 - x = 0$, the potential in Eq. (43) becomes the cubic $\Gamma_0(y) = -\langle \widehat{\Phi}(y) \rangle_0 = \frac{1}{3} y^3$ characteristic of the Airy integral. For the case of JT gravity, we can solve the spectral curve Eq. (16) for $x = \arcsin^2(y)$ (absorbing the factors of $2\pi$ for convenience), and the leading order potential becomes[8]

$$\Gamma_0(y) = -2y + 2\sqrt{1 - y^2} \arcsin y + y \arcsin^2 y. \qquad (45)$$

One can check that around $y = 0$, the JT potential can be expanded as $\frac{1}{3} y^3 + \mathcal{O}(y^5)$. However, its behaviour away from the origin is quite different from the Airy potential, as will be discussed in Appendix A.

The reason to introduce the canonically conjugate coordinate $y$ may seem a bit mysterious at this stage. However, from the point of view of JT gravity the Fourier transformed vertex operators are rather natural objects. If we interpret the real part of $x$ as parametrizing the energy space of the boundary Schwarzian theory, it makes sense to interpret the conjugate variable $y$'s real part as a temperature, $\beta$, or boundary length. So we think of the insertion of $\widehat{\psi}(y)$ as creating a D-brane on which arbitrarily many open JT worldsheets can end. Notice the similarity to our identification of the boundary creation operators $Z(\beta)$ as the inverse Laplace transform of $\mathcal{J} = \mathcal{J}(x) dx = \mathcal{J}(z) dz$:

$$Z(\beta) = \int_{c-i\infty}^{c+i\infty} dx \, e^{\beta x} \mathcal{J}(x), \qquad \widehat{\psi}(y) = \int_{c-i\infty}^{c+i\infty} dx \, e^{-yx} e^{\Phi(x)}. \qquad (46)$$

The above intuition is strengthened by the observation in [60] that the Fourier transform maps the Virasoro constraints for correlation functions of vertex operators to the open topological recursion of [61, 62]. Moreover, we will see in the next section that the $y$-variable naturally arises as a matrix *eigenvalue* in the dual flavor matrix theory.

---

[8]This potential has appeared in the literature once before (as far as we know), cf. [59].

## 3.2 fMT representation of the brane correlator

Having introduced the vertex operators $\psi(x)$, $\psi^\dagger(\mathsf{x})$ and their transformation properties under symplectomorphisms, we can go on to study correlation functions of brane/anti-brane pairs,

$$D_n(X) = \left\langle \psi(x_1)\psi^\dagger(\mathsf{x}_1)\cdots\psi(x_n)\psi^\dagger(\mathsf{x}_n) \right\rangle_{\mathsf{KS}}. \tag{47}$$

These are the analogs of determinant/inverse determinant insertions in a random (color) matrix theory, so we expect to reduce their correlation function to a suitable flavor matrix integral whose dimension is set by the number of vertex operator insertions. To demonstrate this, let us invert the Fourier transforms in Eq. (41):

$$\psi(x) = \int_{\mathcal{C}} \frac{dy}{\sqrt{\lambda}}\, e^{xy/\lambda}\, \widehat{\psi}(y)\,, \qquad \psi^\dagger(\mathsf{x}) = \int_{\mathcal{C}'} \frac{d\mathsf{y}}{\sqrt{\lambda}}\, e^{-\mathsf{x}\mathsf{y}/\lambda}\, \widehat{\psi}^\dagger(\mathsf{y})\,. \tag{48}$$

Substituting these symplectic transformations into the correlation function $D_n(X)$ gives

$$\left\langle \psi(x_1)\psi^\dagger(\mathsf{x}_1)\cdots\psi(x_n)\psi^\dagger(\mathsf{x}_n) \right\rangle_{\mathsf{KS}} = \lambda^{-n}\int dY\, e^{\mathrm{str}(XY)/\lambda}\left\langle \widehat{\psi}(y_1)\widehat{\psi}^\dagger(\mathsf{y}_1)\cdots\widehat{\psi}(y_n)\widehat{\psi}^\dagger(\mathsf{y}_n) \right\rangle_{\mathsf{KS}}. \tag{49}$$

Here we have defined $dY = \prod_i dy_i d\mathsf{y}_i$ and collected the exponentials into a supertrace over graded diagonal matrices $X = \mathrm{diag}(x_1,\ldots,x_n|\mathsf{x}_1,\ldots,\mathsf{x}_n)$ and $Y = \mathrm{diag}(y_1,\ldots,y_n|\mathsf{y}_1,\ldots,\mathsf{y}_n)$. As a next step, we use Wick's theorem to write the operator product of the vertex operators $\widehat{\psi}(y) = e^{\widehat{\Phi}(y_i)}$ and $\widehat{\psi}^\dagger(\mathsf{y}) = e^{-\widehat{\Phi}(\mathsf{y}_i)}$ as a single normal-ordered exponential of chiral bosons. As before, we define normal ordering $\{\cdots\}$ by subtracting all singular terms coming from the OPE of the chiral boson $\widehat{\Phi}(y)\widehat{\Phi}(y') \sim \log(y - y') + \mathrm{reg}$. After doing all the Wick contractions, this procedure gives rise to a super-Vandermonde determinant

$$\mathsf{s}\Delta(Y) \equiv \frac{\prod_{i<j}(y_i - y_j)\prod_{k<l}(\mathsf{y}_k - \mathsf{y}_l)}{\prod_{i,k}(y_i - \mathsf{y}_k)}\,. \tag{50}$$

For example, when $n = 2$, the six possible contractions give the multiplicative factor

$$e^{\log(y_1-y_2)+\log(\mathsf{y}_1-\mathsf{y}_2)-\log(y_1-\mathsf{y}_1)-\log(y_2-\mathsf{y}_2)-\log(y_1-\mathsf{y}_2)-\log(y_2-\mathsf{y}_1)} = \mathsf{s}\Delta(Y)\,. \tag{51}$$

Using Cauchy's determinant formula, the super-Vandermonde determinant can be written more elegantly as

$$\mathsf{s}\Delta(Y) = \det_{ij}\frac{1}{y_i - \mathsf{y}_j}\,. \tag{52}$$

Hence, the brane/anti-brane correlator Eq. (49) can be brought in the following form

$$D_n(X) = \lambda^{-n}\int dY\, e^{\mathrm{str}(XY)/\lambda}\, \mathsf{s}\Delta(Y)\left\langle \left\{ e^{\widehat{\Phi}(y_1)-\widehat{\Phi}(\mathsf{y}_1)+\cdots+\widehat{\Phi}(y_n)-\widehat{\Phi}(\mathsf{y}_n)} \right\} \right\rangle_{\mathsf{KS}}. \tag{53}$$

Using the general formula $\langle \exp\mathcal{O} \rangle = \exp\sum_{k=1}^\infty \frac{1}{k!}\langle\mathcal{O}^k\rangle^{\mathsf{c}}$ for going between correlation functions and connected correlation functions, we represent the brane/anti-brane correlator Eq. (53) in a form that closely resembles a flavor matrix integral

$$D_n(X) = \lambda^{-n}\int dY\, \mathsf{s}\Delta(Y) e^{-\Gamma(Y)/\lambda+\mathrm{str}(XY)/\lambda}\,, \tag{54}$$

where the potential $\Gamma(Y)$ is defined as a sum of connected correlation functions in KS theory

$$\Gamma(Y) = -\sum_{k=1}^\infty \frac{\lambda}{k!}\left\langle \left\{ \left(\widehat{\Phi}(y_1)-\widehat{\Phi}(\mathsf{y}_1)+\cdots+\widehat{\Phi}(y_n)-\widehat{\Phi}(\mathsf{y}_n)\right)^k \right\} \right\rangle_{\mathsf{KS}}^{\mathsf{c}}. \tag{55}$$

As a last step, we normal order the brane correlator once more, but this time in the $(x_i, \mathsf{x}_i)$-coordinates, which gives another super-Vandermonde determinant. We find

$$\left\langle \left\{ \psi(x_1)\psi^\dagger(\mathsf{x}_1)\cdots\psi(x_n)\psi^\dagger(\mathsf{x}_n) \right\} \right\rangle_{\mathsf{KS}} = \frac{1}{\mathsf{s}\Delta(X)}\left\langle \psi(x_1)\psi^\dagger(\mathsf{x}_1)\cdots\psi(x_n)\psi^\dagger(\mathsf{x}_n) \right\rangle_{\mathsf{KS}} \quad (56)$$

$$= \frac{\lambda^{-n}}{\mathsf{s}\Delta(X)} \int dY\, \mathsf{s}\Delta(Y)\, e^{-\Gamma(Y)/\lambda + \mathrm{str}(XY)/\lambda}. \quad (57)$$

This is precisely the eigenvalue representation of a GL$(n|n)$ graded flavor matrix integral with invariant potential $\Gamma(A)$ and 'external source' $X$. To recognize this, consider a Hermitian supermatrix $A$ in GL$(n|n)$ with eigenvalues $\{y_i, \mathsf{y}_i\}$, diagonalized by a unitary supermatrix $T \in \mathsf{U}_f = \mathsf{U}(n|n)$:

$$A = TYT^{-1}, \qquad Y = \mathrm{diag}(y_1,\ldots,y_n | \mathsf{y}_1,\ldots,\mathsf{y}_n). \quad (58)$$

In terms of this decomposition the integration measure $dA$ decomposes as

$$dA = dT\, dY\, \mathsf{s}\Delta(Y)^2, \quad (59)$$

where we have defined $dY = \prod_{a=1}^n dy_a\, d\mathsf{y}_a$, and $dT$ is the Haar measure on U$(n|n)$. Similar to ordinary matrix integrals, the super-Vandermonde determinant arises as the Jacobian of the change of variables from $A$ to $T$ and $Y$. It can be easily derived as the volume form corresponding to the metric on the space of Hermitian supermatrices

$$ds^2 = \mathrm{str}(dA^2) = \mathrm{str}(dY^2 + [d\Omega, Y]^2), \quad (60)$$

where $d\Omega = T^{-1}dT$. Now consider a general flavor matrix integral with invariant potential $\Gamma(A) = \Gamma(TYT^{-1}) = \Gamma(Y)$ and external source term $\mathrm{str}(XA)$. One can see this as the supersymmetric generalization of the Kontsevich matrix integral, whose potential is $\Gamma(A) = \frac{1}{3}\mathrm{str}(A^3)$. However, we will keep $\Gamma(A)$ arbitrary for now, and match it to JT gravity later. We also include a coupling constant $\lambda$. Using the eigenvalue decomposition Eq. (58) the flavor matrix integral decomposes as

$$\int_{(n|n)} dA\, e^{-\Gamma(A)/\lambda + \mathrm{str}(XA)/\lambda} = \int dY e^{-\Gamma(Y)/\lambda}\, \mathsf{s}\Delta(Y)^2 \int_{U(n|n)} dT\, e^{\mathrm{str}(XTYT^{-1})/\lambda}. \quad (61)$$

The super-unitary integral appearing in Eq. (61) can be evaluated in closed form [63,64] and is a supersymmetric generalization of the famous Harish-Chandra-Itzykson-Zuber integral. The integral turns out to be one-loop exact and evaluates to[9]

$$\int_{U(n|n)} dT\, \exp\left[\frac{1}{\lambda}\mathrm{str}(XTYT^{-1})\right] = C_n\, \lambda^{-n} \frac{\det_{i,j}\left(e^{x_i y_j/\lambda}\right)\det_{k,l}\left(e^{-\mathsf{x}_k \mathsf{y}_l/\lambda}\right)}{\mathsf{s}\Delta(X)\mathsf{s}\Delta(Y)}. \quad (62)$$

Plugging this into Eq. (61), we can evaluate the determinants by exploiting the antisymmetry of the Vandermonde determinants $\Delta(y) = \prod_{j<i}(y_i - y_j)$ and $\Delta(\mathsf{y}) = \prod_{j<i}(\mathsf{y}_i - \mathsf{y}_j)$ and the freedom to relabel dummy variables in the $y$ and $\mathsf{y}$ integration. For example, when $n = 2$, there is a factor $-\Delta(y)e^{(x_1 y_2 + x_2 y_1)/\lambda}$, which upon relabeling $y_2 \leftrightarrow y_1$ becomes $+\Delta(y)e^{(x_1 y_1 + x_2 y_2)/\lambda}$. So we are left with only the diagonal contributions of $e^{x_i y_i/\lambda}$ and $e^{-\mathsf{x}_k \mathsf{y}_k/\lambda}$, which assemble

---

[9]This is proven in [63] using a heat kernel method for the super-Laplacian operator on the space of Hermitian supermatrices, analogous to Itzykson and Zuber's original proof [65] in the non-supersymmetric case.

into the supertrace of $XY$. This argument is easily extended to general $n$, and we arrive at the eigenvalue representation of the flavor matrix integral

$$\int_{(n|n)} dA\, e^{-\Gamma(A)/\lambda + \text{str}(XA)/\lambda} = \tilde{C}_n \frac{\lambda^{-n}}{\text{s}\Delta(X)} \int dY\, \text{s}\Delta(Y)\, e^{-\Gamma(Y)/\lambda + \text{str}(XY)/\lambda}. \tag{63}$$

The prefactor $\tilde{C}_n$ (which now includes the symmetry factors from the above permutation argument) can be absorbed in an overall normalization of the flavor matrix integral. If we now identify the fMT coupling constant $\lambda$ with the KS coupling constant, and take as our invariant potential the KS potential Eq. (55), then we see that the flavor matrix integral coincides with the brane/anti-brane correlator Eq. (57). In conclusion, we have shown that the normal ordered expectation value of $n$ brane and $n$ anti-brane creation operators in KS theory, inserted at positions $x_i, \mathsf{x}_i$, is exactly equal to a $(n|n)$ graded flavor matrix integral with external source $X = \text{diag}(x_1, \ldots, x_n | \mathsf{x}_1, \ldots, \mathsf{x}_n)$:

$$\left\langle \left\{ e^{\Phi(x_1)} e^{-\Phi(\mathsf{x}_1)} \cdots e^{\Phi(x_n)} e^{-\Phi(\mathsf{x}_n)} \right\} \right\rangle_{\text{KS}} = \int_{(n|n)} dA\, \exp\left[ -e^{S_0}\Gamma(A) + e^{S_0}\text{str}(XA) \right]. \tag{64}$$

This is the main result of this section. In principle, one can compute the potential $\Gamma(Y)$ to any order in the KS perturbation theory in powers of $\lambda = e^{-S_0}$. In the case that the spectral curve is the Airy curve $y^2 - x = 0$, the higher genus contributions vanish and the only non-zero contributions to $\Gamma(Y)$ come from the disk and cylinder amplitudes.[10] For the JT spectral curve, there are non-trivial corrections suppressed in powers of $\lambda$ (for a single brane insertion these were computed in [59]). In the matrix theory context, such refinement would describe small corrections to the *average* spectral density. However, we are interested in the limit that $e^{S_0}$ is very large, $\Delta E$ very small, and the ratio $s \sim e^{S_0}\Delta E$ kept fixed, as explained in Section 2.1. In this limit, it suffices to keep only the leading order potential function

$$\Gamma(Y) \approx -\sum_{i=1}^{n} \left( \langle \widehat{\Phi}(y_i) \rangle_0 - \langle \widehat{\Phi}(\mathsf{y}_i) \rangle_0 \right) = \sum_{i=1}^{n} \left( \int^{y_i} x(y)dy - \int^{\mathsf{y}_i} x(y)dy \right). \tag{65}$$

As explained before, the function $x(y)$ follows from the spectral curve equation $H(x,y) = 0$, and is given by $x(y) = \arcsin^2(y)$ for the JT spectral curve. And, as advertised, the right-hand side of Eq. (65) can be written as a supertrace, $\Gamma(Y) \approx \text{str}\,\Gamma_0(Y)$, where the function $\Gamma_0(y)$ for the JT spectral curve was given in Eq. (45). Note that for small values of the argument, $x(y) \sim y^2$, and so $\Gamma_0(Y) \sim \text{str}(Y^3)$. Hence, near the spectral edge $E \to 0$ our flavor matrix model is governed by a cubic potential, and thus reduces to graded variant of a Kontsevich matrix model.[11] However, the potential behaves differently at infinity. This will influence the choice of integration contours for the eigenvalues $(y_i, \mathsf{y}_i)$, as will be discussed in more depth in Appendix A.

## 3.3 Reduction to the nonlinear $\sigma$-model

Having derived a flavor matrix integral from (anti-)brane insertions in KS theory, we go on to show that the nonlinear $\sigma$-model introduced in Eq. (14) arises from a stationary phase analysis

---

[10]One way to see this is to note that under the $x$-$y$ symmetry the dual of the Airy spectral curve has no branch point, since $dy(z) = dz$. So the topological recursion of [57] is identically zero. The claim can also be checked order by order, by computing the 'WKB' form of $\langle \psi(x) \rangle = e^{\frac{1}{\lambda}\sum_n \lambda^n S_n(x)}$ using topological recursion [50], and then doing a stationary phase analysis of the Fourier transform in Eq. (41).

[11]This is similar to how the insertion of $N$ FZZT branes in Liouville theory were shown to give rise to a Kontsevich matrix integral by Gaiotto and Rastelli in [66]. Including anti-FZZT branes [67], one expects to find the graded variant of the Kontsevich matrix integral.

of the flavor matrix integral Eq. (64) in the limit of large $e^{S_0}$. More precisely, we study the fMT in the limit discussed in the introduction, where we take $\lambda \to 0, \Delta E \to 0$ and $s \sim \Delta E/\lambda$ held fixed. We will find that which saddles dominate depends on the causal symmetry breaking parameters $\pm i\eta$, which are the infinitesimal imaginary offsets in the (anti-)brane positions $\text{Im}(x_i) = \pm i\eta$ on the spectral curve.

To find the stationary points, we decompose $A = TYT^{-1}$ as before and vary $T$ and $Y$:

$$\text{str}\left[\left(T^{-1}XT + \Gamma_0'(Y)\right)\delta Y\right] = 0,$$
$$\text{str}\left[(YTX - XTY)\delta T\right] = 0.$$

(66)

Let us first discuss the diagonal solutions, for which $T = \mathbb{1}$. In that case, the entries $y_i, \mathsf{y}_i$ of $Y$ separately have to satisfy the equations

$$\Gamma_0'(y_i) = x_i, \quad \Gamma_0'(\mathsf{y}_i) = \mathsf{x}_i.$$

(67)

Looking at the form of $\Gamma_0(y)$ in Eq. (65), these equations are solved by inverting $x(y_i) = x_i$, where $x(y)$ is determined by the spectral curve Eq. (16). Here the branched structure of the spectral curve rears its head: for each diagonal element $y_i, \mathsf{y}_i$, there are two choices of branch when taking the square root, for example

$$y_1^\pm = \pm \sin(\sqrt{x_1}) = \pm i\rho_0(E_1).$$

(68)

This naively gives a total of $2^n$ diagonal saddles in the fMT. However, precisely which saddle points contribute depends on the integration contour that is part of the definition of the flavor matrix integral. In Appendix A we perform a steepest descent analysis of the fMT with the potential $\Gamma_0(Y)$. We find that the dominant saddle is selected by the $\pm i\eta$ prescription of the external energy arguments. For example, if we take the imaginary part of $X$ to be $i\eta\,\hat{\tau}_3$, where $\hat{\tau}_3 = \tau_3 \otimes \mathbb{1}$ is the (flavor) Pauli $z$ matrix, then the dominant saddle point will be

$$Y_{st} = i\rho_0(E)\,\tau_3 \otimes \mathbb{1}.$$

(69)

This is called the *standard saddle*. We have set the external energies equal to $E$ in $Y_{st}$, because upon evaluating the potential at the saddle point the linear term $e^{S_0}\text{str}XY_{st}$ should be expanded to leading order in $s \sim e^{S_0}\Delta E$. Besides the standard saddle, there are also subleading *Andreev-Altshuler* (AA) saddles [28], which arise from $i\eta$-prescriptions such that a brane and an anti-brane are on opposite sheets before taking their OPE limit. For example, in computing the density-density correlator ($n = 2$), there is one such AA saddle, given by

$$Y_{AA} = i\rho_0(E)\,\tau_3 \otimes \tau_3.$$

(70)

It can easily be verified that evaluating the fMT on the standard saddle gives a vanishing action, while the action for the AA saddle is non-zero and purely imaginary. This gives rise to the well-known oscillatory behavior of the spectral density two-point function (the 'sine kernel') [68].

Having found the diagonal saddle points $Y_*$, we make the following simple observation: if $[X, T] = 0$, then the saddle point equations Eq. (66) are also solved by $TY_*T^{-1}$. For generic values of the external energies, $X$ does not commute with $T$. However, recall that we are considering the limit that $\Delta E$ is very small, and so we can approximate $[X, T] \approx 0$. In other words, to leading order in $s$, $TY_*T^{-1}$ is an *approximate* solution to the saddle point equations. So instead of a sum over distinct saddles, we have to integrate over a whole saddle point manifold.

To determine the saddle point manifold, we need to account for the redundancies corresponding to transformations that commute with the diagonal saddle $Y_*$. Without loss of

generality, consider for $Y_*$ the standard saddle, $Y_{st}$, which is proportional to $\hat{\tau}_3$. The stabilizer subgroup of $Y_{st}$ is $U(\frac{n}{2}|\frac{n}{2}) \times U(\frac{n}{2}|\frac{n}{2})$, which acts on $U(n|n)$ in an obvious way. So the full saddle point manifold will be the coset manifold

$$\text{AIII}_{n|n} = \frac{U(n|n)}{U(\frac{n}{2}|\frac{n}{2}) \times U(\frac{n}{2}|\frac{n}{2})} \tag{71}$$

as advertised in Section 2.1. The saddle point manifold continuously connects the standard saddle to the AA saddles [18]. Parametrizing the coset by $Q = T\hat{\tau}_3 T^{-1}$, and evaluating the fMT action on its solution $A = i\rho_0(E)Q$, we obtain the non-linear $\sigma$-model on the Goldstone manifold

$$\int_{(n|n)} dA \exp\left[-e^{S_0}\Gamma(A) + e^{S_0}\text{str}(XA)\right] \simeq \mathcal{N} \int_{\text{AIII}_{n|n}} dQ \exp\left[i\frac{\rho_0(E)}{\lambda}\text{str}(XQ)\right], \tag{72}$$

to first order in $s$, in the late time limit $\lambda \to 0$, $\Delta E \to 0$. In the above expression we have absorbed the potential term $\exp\left[-e^{S_0}\text{str}\,\Gamma_0(Y)\right]$ into a normalization $\mathcal{N}$, because it is independent of $Q$ using the cyclicity of the supertrace. Moreover, since $Q$ is supertraceless, we can freely replace $X$ by its symmetry breaking part $X \to X - E\mathbf{1}$ containing the energy differences $\Delta E$ only. Concluding, we see that the brane/anti-brane correlator in KS theory captures the universal late time ergodic physics described by the NLSM.

Having derived the non-linear $\sigma$-model of quantum chaos from KS theory, one can systematically study perturbative corrections in the parameter $s$. As shown in [18], the $s^{-1}$-expansion of spectral correlations is a topological expansion. This expansion should be viewed as a limit of the JT topological expansion for which the probe arguments are sent to small differences, $\Delta E$, and only the relevant diagrams are kept. This can checked by explicit computation: Individual diagrams contributing to the expansion of the NLSM can be represented in a 't Hooft double line syntax [18,19] to verify that their perturbative $s^{-1}$-degree maps to the topological order of the KS/JT expansion. Conversely, one may take the limit $E_i \to E_j$ in contributions of given genus order in the expansion of the JT path integral to verify that the topological order determines the order of the highest singularities in $s^{-1} \propto |E_i - E_j|^{-1}$, with matching coefficients.

However, we already mentioned that the correspondence between the asymptotic expansion of the JT path integral in $e^{-S_0}$ to the NLSM is limited to values $s^{-1} < 1$. In the opposite case, corresponding to post-Heisenberg or plateau times, the NLSM leaves the regime of perturbation theory. Instead, the now small coupling constant requires non-perturbative integration over the full graded coset manifold. This integration, which describes the restoration of the causal symmetry previously broken by large fluctuations, has no analog in semi-classical JT gravity. We conclude that the latter knows about perturbative signatures of level correlations, but not about their fine-grained microscopics. In this sense, JT gravity remains 'UV incomplete'. Our discussion has shown that the closure of the theory is provided by KS field theory, which for pre-Heisenberg time scales is perturbatively equivalent to JT, and to the NLSM beyond.

In this context, it is also worth mentioning connections to the SYK model. Early work identified a bridge between SYK and JT at time scales which from the perspective of our present discussion are 'super-short', $t \sim \log(e^{S_0})$. In this regime, reflecting the approximate realization of a conformal symmetry, both reduce to Liouville quantum mechanics [69] as a common effective theory. At larger scales beyond the Thouless time (here identified with the dip time of the SYK form factor), the SYK model is described by fMT [43], which close to the lower spectral edge again takes the form of a graded Kontsevich model [18]. In this way a second link between the SYK model (at the spectral edge) and a gravitational theory is drawn, via universe

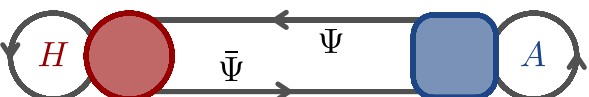

Figure 3: A schematic picture of the different types of D-branes in KS theory. Color branes (red) and flavor branes (blue) each have open string degrees of freedom associated to the branes themselves (indicated by the color matrix $H$ and flavor matrix $A$ resp.) and open string degrees of freedom $\Psi, \overline{\Psi}$ connecting both types of branes.

field theory.[12] This 'late time bridge' relies on conceptually independent insights to that at early times. Its underlying symmetry principle is the causal symmetry breaking/restoration of ergodic quantum systems. Understanding how the respective symmetry principles connect at intermediate time scales remains an open question.

## 4 D-branes and the color-flavor map

In the previous section we have shown how fMT arises directly from universe field theory. This has been derived solely using a closed string field theory framework. However, since a crucial role is played by D-branes in the target space geometry, it is natural to expect that there is also an open string field theory which leads to fMT. In this section we will show that this is indeed the case, owing to an open-closed duality in string field theory defined on a particular 6d Calabi-Yau. This open-closed duality manifested itself in the previous section – in the 2d target space theory on the spectral curve – as a type of boson-fermion correspondence between 'fermionic' operators $\psi(x)$ and vertex operators $e^{\Phi(x)}$. In the six-dimensional setting, the open-closed duality relates Kodaira-Spencer theory (closed) to holomorphic Chern-Simons theory (open) [46]. Reducing this holomorphic Chern-Simons theory to the worldvolume of a stack of non-compact branes and anti-branes gives rise to the fMT, as will be explained in Section 4.1.

Moreover, the open string perspective naturally explains the color-flavor duality described in Section 2.1. Recall from that section that the long time limit of ergodic dynamics realized in Hilbert spaces of high color dimension, $L$, is equivalently described by a matrix theory of low flavor dimension. Here we discuss a gravitational interpretation of the color and flavor degrees of freedom, and describe their duality from a D-brane worldvolume perspective. An idea of the relevant constructions is given in Fig. 3.

Before describing these D-branes explicitly, let us start with the general picture to have in mind. As always, let us begin from the determinant ratio Eq. (1). We can rewrite it in terms of graded integrals, keeping in mind the interpretation of branes for the determinants in the numerator and anti-branes for those in the denominator. We thus have

$$\mathrm{Sdet}(X \otimes \mathbb{1}_{\mathrm{c}} + \mathbb{1}_{\mathrm{f}} \otimes H) = \int D\bar{\Psi} D\Psi \exp\left[-\bar{\Psi}\left(X \otimes \mathbb{1}_{\mathrm{c}} + \mathbb{1}_{\mathrm{f}} \otimes H\right)\Psi\right], \tag{73}$$

where $\Psi$ is a graded vector of dimension $(nL|nL)$. The adjoint flavor group representation carried by the operator $\Xi = X \otimes \mathbb{1}_{\mathrm{c}} + \mathbb{1}_{\mathrm{f}} \otimes H$ in Eq. (10) implies the following transformation in the fundamental representation for the vectors $\Psi$ and $\overline{\Psi}$

$$\Psi \to T\Psi, \qquad \overline{\Psi} \to \overline{\Psi} T^{-1}, \qquad T \in \mathrm{U}(n|n). \tag{74}$$

---

[12]It would be interesting to see how our work relates to another recent connection between SYK and string theory established in Ref. [70].

The idea is now to take the integral representation in Eq. (73) seriously, and identify the vector $\Psi$ as describing the open string degrees of freedom that stretch between two types of branes in KS theory: non-compact branes (which we already touched upon in Section 2.2) and compact branes. The causal symmetry given in Eq. (74) can now be identified with the gauge symmetry of the gauge field on the configuration of $n$ coincident branes and anti-branes. Crucially, the Hermitian color matrix $H$, which describes the microscopic degrees of freedom, is taken to large size $L \to \infty$ where $L$ counts the number of compact branes. However, the flavor matrix $A$ has finite (possibly small) size $2n$, where the index $n$ counts the number of non-compact (anti-)branes.

As explained in Section 2.2, the target space geometry is the non-compact Calabi-Yau

$$\text{CY}: \quad uv - H(x, y) = 0, \tag{75}$$

which is a fibration over the spectral curve defined by $\mathcal{S} : H(x, y) = 0$. One recovers JT gravity for the choice $\mathcal{S} = \mathcal{S}_{\text{JT}}$. There are now two distinct ways of introducing D-branes in the geometry, cf. Ref. [31]: we can either introduce non-compact 'flavor' branes, which can be related to a Kontsevich-like matrix model (see Section 4.1) or compact 'color' branes, which lead to the usual Hermitian matrix model (see Section 4.2). The color branes are wrapped over a compact two-cycle in the CY geometry and their presence introduces a flux for the holomorphic $(3,0)$-form

$$\Omega = \frac{du}{u} \wedge dx \wedge dy, \tag{76}$$

over the three-sphere linking the two-cycle. One can also deform the complex structure at infinity by inserting flavor branes which wrap the non-compact fibers $u = 0, v = 0$ in the CY. The open string sector associated to the branes can be interpreted as an effective change of the geometry in which the closed strings propagate. In both cases the world-volume theory associated to the branes can be derived from the dimensional reduction of a holomorphic CS theory on the space-filling D6 brane wrapping the entire CY [71]

$$S_{\text{open}} = \frac{1}{g_s} \int_{\text{CY}} \Omega \wedge \text{str}\left[ \mathcal{A} \wedge \bar{\partial} \mathcal{A} + \frac{2}{3} \mathcal{A} \wedge \mathcal{A} \wedge \mathcal{A} \right], \tag{77}$$

where $g_s$ is the (open-)string coupling.[13] The supertrace str indicates that we are considering a slightly unconventional version of the open string field theory, that includes both branes and anti-branes (which have opposite flux). It was argued in [37] that the inclusion of anti-branes can be implemented by upgrading the $(0,1)$-form gauge field $\mathcal{A}$ to be supermatrix-valued. In order to describe, say, a stack of $n$ branes and $n$ anti-branes wrapped on $\mathcal{B}$, we take the gauge group in Eq. (77) to be the supergroup $\text{GL}(n|n)$. This can be argued for topological branes by examining their Chan-Paton factors [37], which carry opposite signs for string world-sheets with an odd number of boundaries on an anti-brane. By examining the four different kinds of annulus diagrams that correspond to the string having endpoints on either a brane or an anti-brane, and assigning a minus sign to each anti-brane boundary, one sees that the physical states arrange themselves into a $\text{U}(n|n)$ superconnection, or rather its complexification $\text{GL}(n|n)$. We therefore end up with a $\text{GL}(n|n)$-valued version of the holomorphic Chern-Simons theory describing the B-model on the CY Eq. (77).

## 4.1 Non-compact branes: Flavor

Let us first define the relevant probe flavor branes in the geometry Eq. (75). For a fixed point $(x_0, y_0)$ in the $(x, y)$-plane the equation

$$uv = H(x_0, y_0) \tag{78}$$

---

[13]For the reader who may want to compare with the matrix-model convention frequently employed in the literature, comparing the Dijkgraaf-Vafa matrix potential (94) with equation (3.9) in [18], reveals the relation $g^2 = g_s L$.

defines a subspace of complex dimension one. When $(x_0, y_0) \in \mathcal{S}$, the above curve develops a node $uv = 0$ and splits in two complex planes: $u = 0$ or $v = 0$. These planes are wrapped by the flavor branes. Let us assume that the branes are at some fixed position

$$\mathcal{B}: \quad u = 0, \qquad (x, y) = (x_0, y_0), \tag{79}$$

where $(x_0, y_0) \in \mathcal{S}$ lies on the spectral curve. The world-volume of the brane – or as will be the case relevant to us, a stack of branes – is parametrized by the complex coordinate $(v, \bar{v})$. We may also wrap anti-branes along these fibres, the only difference being their orientation which is opposite to that of the branes.[14] In the following, we will consider normal deformations of the brane Eq. (79) with suitable boundary conditions at infinity $|v| \to \infty$.

We now want to describe the effective theory associated to the open strings ending on $\mathcal{B}$. We will show that the result takes the form of a flavor matrix model. This derivation closely follows [31,72] (with the exception that we accommodate both branes and anti-branes). The dimensional reduction of the holomorphic CS action Eq. (77) involves the following decomposition: the gauge field $\mathcal{A}$ splits as a gauge field $\tilde{\mathcal{A}}$ on the world-volume of the brane and two Higgs fields $A$ and $B$, which describe the movement of the brane in the transverse direction. We will assume that all fields depend only on the variables $(v, \bar{v})$ along the brane $\mathcal{B}$, so that we can apply a dimensional reduction. Explicitly, the normal deformations of the brane Eq. (79) are given by two scalar fields $B = B(v, \bar{v})$ and $A = A(v, \bar{v})$ in the adjoint representation of GL$(n|n)$. They represent the deformations of $\mathcal{B}$ in the directions of the $(x, y)$-plane, according to the identification

$$(x, y) \quad \mapsto \quad (B(v, \bar{v}), A(v, \bar{v})) . \tag{80}$$

Moreover, the holormorphic $(3, 0)$-form Eq. (76) can be written as

$$\Omega = -\frac{dv}{v} \wedge dx \wedge dy, \tag{81}$$

in terms of the variable $v$. This shows that only non-zero contributions in Eq. (77) come from the $d\bar{v}, d\bar{x}$ and $d\bar{y}$ components of the gauge field $\mathcal{A}$, which we have identified with the fields $\tilde{\mathcal{A}}(v, \bar{v}), B(v, \bar{v})$ and $A(v, \bar{v})$ respectively. The dimensional reduction now leads to the action [72, 73]

$$S_{\text{flavor}} = -\frac{1}{\lambda} \int_{\mathcal{B}} \frac{i}{2} \frac{dv \, d\bar{v}}{v} \, \text{str}\left[ B \overline{D} A \right], \tag{82}$$

where $\overline{D} = \bar{\partial} + [\tilde{\mathcal{A}}, \cdot]$ is the antiholomorphic covariant derivative associated to $\tilde{\mathcal{A}}$ and $\bar{\partial}$ is the derivative with respect to $\bar{v}$. The volume of the transverse directions that have been integrated out have been absorbed into the coupling constant,[15] which we identify with the expansion parameter $\lambda$, since we are considering a stack of branes in the (double-scaled) closed string background. Integrating the holomorphic three-form $\Omega$ yields a factor of $L g_s$, measuring the flux of the $L$ compact branes that have been dissolved in the geometry (see Section 4.3 below for more detail), and we needed to rescale $A \to A/g_s^{1/4}$, $B \to B/g_s^{1/2}$ to get a finite answer in the double-scaling limit, $L \to \infty, g_s \to \infty$ with $1/\lambda = e^{S_0} = \left(L/g_s^3\right)^{1/4}$ held finite.

Note that this theory inherits a GL$(n|n)$ gauge symmetry, which acts as

$$B \mapsto g B g^{\dagger}, \qquad A \mapsto g A g^{\dagger}, \qquad \tilde{\mathcal{A}} \mapsto g \tilde{\mathcal{A}} g^{\dagger} + (\bar{\partial} g) g^{\dagger}, \tag{83}$$

where $g(v, \bar{v})$ is a GL$(n|n)$ transformation depending on the fibre coordinate on $\mathcal{B}$. Gauge fixing to $\tilde{\mathcal{A}} = 0$ leaves only the constant $g \in$ GL$(n|n)$, which for $n = 2$, we recognise as the causal

---

[14]Equivalently, one can obtain an anti-brane insertion on the spectral curve by putting a brane along the $v = 0$ direction (instead of $u = 0$). This follows from Eq. (79), which implies $\frac{du}{u} = -\frac{dv}{v}$, and so in terms of $v$ the holomorphic $(3, 0)$-form Eq. (76) acquires a minus sign.

[15]A compactification of the transverse directions is necessary to make the volume finite.

symmetry transformations appropriate for a four-determinant ratio. In fact, we are interested in configurations of the branes such that their classical position is fixed to the diagonal matrix $X$ of external energies introduced in Eq. (1). We see here that this corresponds to fixing the asymptotic positions of the flavor branes on the spectral curve. These boundary conditions can be implemented by a adding a boundary term (which we can think of as a Legendre transform)

$$S_{\text{flavor}} = \frac{1}{\lambda} \int_{\mathcal{B}} \frac{i}{2} \frac{dv d\bar{v}}{v} \, \text{str} (B - X) \overline{D} A, \tag{84}$$

implying that the field $B$ at infinity is now fixed to the constant matrix $B_\infty = X$, while $A_\infty$ at infinity is free, and must be integrated over in the quantum theory. As we described before, if the boundary condition parametrized by $X$ is not proportional to the identity matrix in flavor space, i.e. contains unequal energy arguments on the diagonal, the small differences break the causal symmetry explicitly. This action will give rise to a Kontsevich-like matrix model for the flavor degrees of freedom in terms of the matrix $A_\infty$.

To see this, let us study the action Eq. (82) in a bit more detail. First, the gauge field $\tilde{A}$ can be set to zero by a suitable gauge transformation. The equation of motion for $\tilde{A}$, given by $[A, B] = 0$, has to be imposed as a constraint, implying that the matrices associated to the Higgs fields can be simultaneously diagonalized. To simplify our analysis, we assume that the functions $B$ and $A$ are rotationally symmetric and only depend on the radial direction $r \equiv |v|$ of the brane. One can now perform the integral over the angular coordinate $\theta \equiv \arg v$. This gives an extra factor of $2\pi$ and leaves an integral over the radial direction:

$$S_{\text{flavor}} = \frac{2\pi}{\lambda} \text{str} \left[ \int_0^\infty dr \, (B - X) \partial_r A \right] = \frac{2\pi}{\lambda} \text{str} \left[ \int_{A_0}^{A_\infty} (B - X) \, dA \right]. \tag{85}$$

Recall that we have assumed that the brane is at some fixed position $B_\infty = X$ at infinity $v \to \infty$, and takes some possibly different value on the spectral curve at the origin $v = 0$. This leads to the final form of the effective action (up to an irrelevant constant):

$$S_{\text{flavor}} = e^{S_0} \text{str} \left[ \int^{A_\infty} B dA - X A_\infty \right], \tag{86}$$

where we have used that $\lambda = e^{-S_0}$. Note that the equation of motion for this action is given by $B = X$, so we can indeed interpret the matrix $X$ as describing the classical position of the branes in the $x$−plane.

In the case that the spectral curve $\mathcal{S}$ is given by $y^2 - x = 0$ (i.e. pure topological gravity), the action takes a familiar form. The equation $B = B(A)$ can be solved directly as a function of $A$, and by integrating Eq. (86) we obtain:

$$S_{\text{flavor}} = e^{S_0} \text{str} \left[ \frac{1}{3} A^3 - X A \right], \tag{87}$$

where we have renamed – by a slight abuse of notation – $A \equiv A_\infty$ to conform with the notation that was used before. This is a graded version of the Kontsevich model action with cubic interaction. Since we have Legendre transformed to an open boundary condition on $A(v, \bar{v})$, in the quantum theory we still need to integrate over the value $A_\infty$, which has been identified with the matrix field $A$. The partition function associated to the flavor branes is therefore given by

$$Z_{\text{flavor}}(X) = \int dA \exp \left[ -e^{S_0} \text{str} \left( \frac{1}{3} A^3 - X A \right) \right], \tag{88}$$

up to some overall normalization. For this reason, the action in Eq. (86) gives rise to a graded Kontsevich model. In fact, we can easily generalize this to a general spectral curve $H(x, y) = 0$. In this case the resulting fMT takes the form

$$Z_{\text{flavor}}(X) = \exp\left[-e^{S_0}\,\text{str}\,(\Gamma_0(A) - XA)\right], \qquad \text{with} \qquad \frac{\delta\Gamma_0(A)}{\delta A} = B(A), \qquad (89)$$

where $B(A)$ is determined by the equation $H(B, A) = 0$. It will not have escaped the reader's attention that this is effectively the flavor matrix integral Eq. (64), in the large $e^{S_0}$ limit. This establishes a representation of the fMT (as found in Section 3.2) in a gravitational setting, as a theory of non-compact flavor (anti-)branes probing the backreacted closed-string background.

The integration over $A$ in the quantum theory represents an integration over the position of the brane at infinity. In the semiclassical limit $e^{S_0} \gg 1$, where we also take the energy arguments in $X$ close together, we may evaluate this matrix integral by the method of steepest-descent, parallel to the analysis of Section 3.3. Among the different possible choices of saddle point configurations of the matrix $A$ two are distinguished by being attainable by a contour deformation from the original contour. We remind the reader that this is explained in detail for both the Airy case and the JT spectral curve in Appendix A. In the present case these correspond to semi-classical configurations of the gauge field $\mathcal{A}_*$, such that the original Chern-Simons GL($n|n$) gauge symmetry, already dimensionally reduced to Eq. (83) is broken to those $g \in$ GL($n|n$) which preserve $\mathcal{A}_*$. In both cases, this corresponds to the breaking

$$\text{GL}(n|n) \to \text{GL}\left(\tfrac{n}{2}\big|\tfrac{n}{2}\right) \times \text{GL}\left(\tfrac{n}{2}\big|\tfrac{n}{2}\right),$$

by the classical configuration of the flavor branes.

By considering the non-compact flavor branes in open string field theory, we identified classical saddle point configurations of the branes which encode the standard and Altshuler-Andreev saddle points. One may morally compare this situation with that of adding flavor degrees of freedom in AdS/QCD [74]: there one places flavor D8-branes into the backreacted geometry of backreacted D4-branes, which have been replaced by the closed string geometry, containing an AdS$_5$ factor. The action of the flavor-branes is the familiar (non-abelian) DBI action. Here we think of the spectral curve $H(x, y)$ as the closed-string geometry, and the flavor branes we place into this background are described by the holomorphic Chern-Simons theory. The important difference to standard AdS/CFT is that in our picture, the flavor branes are objects that arise as boundary conditions on JT (or topological gravity) universes, while in the AdS/QCD context they are boundary conditions on fundamental strings embdedded *inside* the AdS background. In both cases the relevant symmetry (causal or chiral symmetry) is broken by specific semiclassical brane configurations. It may be enlightening to pursue this analogy further.

## 4.2 Compact branes: Color

We will now address the question of how to realize the color matrix $H$ in the KS theory. This will involve the introduction of a set of compact branes in a slightly different geometry, that is related to the closed-string background Eq. (75) by a geometric transition. In particular, we will find that the large $L$ Hermitian matrix integral will give rise to the JT target space geometry, which provides an interesting perspective on how the microscopic degrees of freedom are converted into geometry.

The CY geometry that gives rise to this type of matrix integral takes the form of Eq. (75) with, cf. [29, 30]:

$$H(x, y) = y^2 - V'(x)^2. \qquad (90)$$

The function $V$ is directly related to the potential of the Hermitian matrix integral, $V(H)$. Note that the CY geometry defined by Eq. (90) is singular along the slice $u = v = y = 0$: there are

singularities at the critical points $x_c$ satisfying $V'(x_c) = 0$. It is useful to keep in mind the example $V'(x) = x$, which corresponds to a Gaussian matrix integral and has a single critical point at $x_c = 0$. To make the singularity manifest, we can redefine $u = u' - iv'$, $v = u' + iv'$, $y = iy'$ so that the geometry takes the form $u'^2 + v'^2 + x^2 + y'^2 = 0$, which is the defining equation of a conifold. For a more general choice of potential, the geometry in Eq. (90) looks locally like a conifold near each one of the critical points, as long as $V''(x_c) = 0$.

A well-known procedure for removing such conifold singularities is by 'blowing up' the relevant singular points into finite two-spheres. The resulting geometry is referred to as the resolved CY, for which we write $\mathsf{CY}_{\mathrm{res}}$, and is, locally near the singular point, a fiber bundle over the blown-up $\mathbb{P}^1$. The coordinate on the $\mathbb{P}^1$ is denoted by $z$, while the two fiber directions are represented by sections $\chi, \varphi$ with corresponding transition functions

$$z' = 1/z\,, \qquad \chi' = \chi\,, \qquad \varphi' = z^2 \varphi + V'(\chi)z\,, \tag{91}$$

in going from the patch associated to the north pole (indicated by the coordinate $z$) to the south pole (indicated by the coordinate $z'$). We can rewrite the above transition map in a slightly different way by defining the coordinates

$$x \equiv \chi\,, \qquad u \equiv 2\varphi'\,, \qquad v \equiv 2\varphi\,, \qquad y \equiv 2z'\varphi' - V'(x)\,. \tag{92}$$

Then, Eq. (91) becomes

$$\mathsf{CY}: \quad uv - y^2 + V'(x)^2 = 0\,. \tag{93}$$

Note that this is precisely the geometry given by Eq. (90). The blown-up two-spheres are located at the zeros of $V'$, because this is the only way to have $\varphi = \varphi' = 0$.

The relation of the resolved geometry $\mathsf{CY}_{\mathrm{res}}$ to the $L \times L$ Hermitian matrix integral is through a stack of $L$ topological branes that wrap the blown-up singularity.[16] The effective theory associated to the compact branes can again be extracted from Eq. (77), now defined on the target space Eq. (93). The dimensional reduction involves the movement of the brane in the transverse fiber directions $\chi, \varphi$. Since we are dealing with a configuration of multiple branes, these fields are upgraded to matrices in the adjoint reprentation of the gauge group $\mathrm{U}(L)$. Following a similar procedure as for the non-compact branes in Section 4.1, one finds that the partition function of the color branes localizes to a Dijkgraaf-Vafa matrix integral with potential $V$ (the details of this derivation are nicely worked out in [30,75]):

$$\mathcal{Z}_{\mathrm{color}} = \int dH\, e^{-\frac{1}{g_s}\operatorname{tr} V(H)}\,. \tag{94}$$

The constant $L \times L$ matrix $H$ is identified with the scalar $\chi$. We have also introduced the coupling constant $g_s$ associated to the open string background Eq. (93) (pre-double-scaling). Going from the open-string to the closed-string background involves a double-scaling procedure, while simultaneously zooming in on the edge of the eigenvalue spectrum by taking the 't Hooft parameter $g^2 \equiv g_s L$ large. The definition of the matrix integral in Eq. (94) depends on a choice of contour integral. In particular, one can choose a contour that leads to real eigenvalues, and Eq. (94) then takes the form of an $L \times L$ Hermitian matrix integral. The eigenvalues of $H$ corresponds (using the definition Eq. (92)) to the $x$−position of the branes in the geometry Eq. (93). The classic equation of motion is given by $V'(H) = 0$ and therefore the eigenvalues of $H$ are classically located at the critical points of the potential. This is precisely the configuration of topological branes that we are considering. We conclude that, in our construction, the cMT is realized as the effective theory associated to the compact branes.

---

[16]In the case of more than one singularity, one can divide the branes over the different two-spheres.

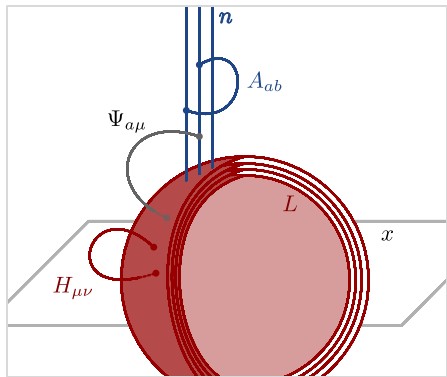 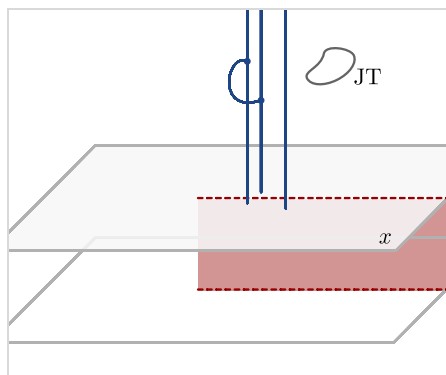

Figure 4: Left: The setup of a stack of $L$ color branes and $n$ flavor (anti-)branes. The open string degrees of freedom that stretch between the two types of branes can be described by a field $\Psi^a_\mu$ in the bi-fundamental representation of $U(L) \otimes U(n|n)$. Right: After taking $L \to \infty$ the color branes dissolve into a flux for the holomorphic three-form, which is represented by a branch cut (dashed line) in the $x$-plane. The closed string description is that of JT gravity on the world-sheet. The flavor branes (blue) in the closed string background, on which JT universes with fixed energy boundaries can end, correspond to probes for the color degrees of freedom.

### 4.3 Color-flavor map and the geometric transition

Having described the color and the flavor perspective, let us now discuss the geometric interpretation of their *duality*, and its relation to the gravitational theory. To describe the multi-determinant operators in Eq. (73) we consider the situation where we have both color and flavor branes in the geometry Eq. (93). The flavor branes introduce a new sector of open strings that stretch between both types of branes. These are described by fields living on the intersection locus (which can be taken to be a single point on the blown-up two-sphere, say at $z = 0$): a bi-fundamental field $\Psi = \Psi^a_\mu$, whose Chan-Paton index $\mu$ labels the $L$-dimensional color Hilbert space, while $a$ labels the flavor Hilbert space. The flavor indices have a $(n|n)$ grading so that $\Psi^a$ contains both fermionic and bosonic components, indicating the presence of both branes and anti-branes. See the left panel of Fig. 4 for an overview of the relevant D-brane configuration.

The idea is now to start from the expression in Eq. (73) involving the open string stretching $\Psi^a_\mu$ between the color and flavor branes, and integrate out the color degrees of freedom. We will now highlight some of the important features of this computation. As was shown in Section 4.2, the theory associated to the compact branes is a Hermitian matrix integral with potential $\operatorname{tr} V(H)$. Therefore, this step involves taking the average with respect to $\langle \dots \rangle_H$. This effectively introduces bound states of open strings (which are the equivalent of 'mesons' in QCD):

$$\Pi^{ab} = \Psi^a_\mu \overline{\Psi}^b_\mu, \tag{95}$$

where the color index is summed over, so that they are indeed color singlets. To be precise, the terms involving the color matrix $H$ are replaced by a potential term $\operatorname{str} V(\Pi)$ for the $\Psi$-fields. One can now integrate in a graded matrix $A^{ab}$ that couples to $\Pi^{ab}$ through $\operatorname{str} A\Pi = \overline{\Psi}(A \otimes \mathbb{1}_c)\Psi$ and effectively replaces the potential by $\operatorname{str} V(A)$. In the target space geometry, the field $A^{ab}$ describes open strings stretching between two flavor branes (as indicated in Fig. 4). The integral over $\Psi$-fields leaves a determinant operator $\operatorname{Sdet}(X \otimes \mathbb{1}_c + A \otimes \mathbb{1}_c)^L$, which realizes the color-flavor duality Eq. (11). After applying a double-scaling limit (where both the size of the color matrix size $L$ and the open string coupling constant $g$ are taken to be large, while their

ratio is kept fixed) to the flavor matrix integral, one precisely lands on the graded Kontsevich model that we found as an effective theory of the flavor branes in Section 4.1.

From the gravitational perspective, we can think about the above discussion in terms of an open/closed duality. By integrating out the color degrees of freedom, and taking the double-scaling limit $L \to \infty$, we are, in fact, replacing the compact branes by a backreacted target space geometry for the closed string. To be precise, the color branes dissolve into a branch cut, which leads to a non-trivial spectral curve $\mathcal{S}$. The theory of closed string propagation in this modified target space geometry is precisely the KS theory. Its mini-universe expansion in different world-sheet topologies gives rise to the universes on which the gravitational theory lives. In the case of $\mathcal{S} = \mathcal{S}_{\text{JT}}$ this description is JT gravity. The open-string geometry in Fig. 4 left is, therefore, to be contrasted with the closed-string geometry in Fig. 4 right. We have effectively replaced the $H$/color-description with a geometric description in terms of JT gravity.

Given that the probe flavor branes are still there after the large $L$ transition (while the color branes have disappeared), we obtain an additional sector in the theory. On the closed string side of the duality, the probe branes introduce another set of open string degrees of freedom, namely of open JT universes with fixed energy boundaries, that stretch between two flavor branes (these are precisely the $A$-fields). The KS theory allows for the inclusion of such degrees of freedom in terms of vertex operators $\psi = e^{\Phi}, \psi^{\dagger} = e^{-\Phi}$. This provides a geometric understanding of the relation Eq. (64) between vertex operator insertions and the fMT that was derived by an explicit computation in Section 3.

Let us end this section with some more details on the relation between the open- and closed-string background geometries. We have seen that there is a way to resolve the singularities in Eq. (93) by inserting a $\mathbb{P}^1$ at each of the singular points, leading to $\text{CY}_{\text{res}}$. There is actually another way of smoothing out the singularities by deforming the complex structure. For the conifold geometry this can be done by turning on a parameter $\mu$ on the right-hand side of the equation: $u'^2 + v'^2 + x^2 + y'^2 = \mu^2$. Having $\mu > 0$ corresponds to inflating a three-sphere of radius $\mu$: the $S^3$ appears as a real section of the conifold. For a general singularity of the form Eq. (93) one needs a polynomial $\mu(x)$ of degree $n-1$ to deform all the singularities

$$\text{CY}_{\text{def}}: \quad uv - y^2 + V'(x)^2 = \mu(x). \tag{96}$$

Near each of the singular points the geometry looks like the deformed conifold. Viewing the inflated $S^3$ as a two-sphere fibered over an interval in the complex $x$−plane, which appears as a branch cut. Indeed, taking again the conifold as example the relevant interval is $-\sqrt{\mu} \leq x \leq \sqrt{\mu}$. For fixed $x$ the deformed conifold describes a two-sphere of radius $\sqrt{\mu^2 - x^2}$, which disappears at the endpoints of the interval: the total space is therefore a three-sphere. Effectively, in going from $\text{CY}_{\text{res}}$ to $\text{CY}_{\text{def}}$, we thus replace each critical point (or 'blown-up' $\mathbb{P}^1$) by a branch cut (or 'inflated' $S^3$). This is precisely what happens in the double-scaling limit of the cMT Eq. (94), where the eigenvalues cluster together around each critical point leading to a branch cut in the spectral $x$−plane.

The above transition between the resolved and deformed conifold is known as the '*conifold transition*' [76,77], and it is the archetypical example of an open/closed duality. The physical interpretation of the open/closed duality (or 'geometric transition') is that the open string theory on $\text{CY}_{\text{res}}$ with $L$ branes wrapping the two-spheres is equivalent in the double-scaling limit $L \to \infty$ to the closed topological string theory on $\text{CY}_{\text{def}}$, without the D-branes. The D-branes have been replaced by fluxes for the holomorphic $(3,0)$-form. The effective theory associated to the open strings is a matrix integral Eq. (94) with potential $V$. The closed string theory, on the other hand, is the KS theory of complex structure deformations of the spectral curve $\mathcal{S}$. In that sense, the KS on $\mathcal{S}$ is dual to the large $L$ matrix integral [29], with $e^{S_0} \sim L^{1/4}/g_s^{3/4}$. As we have mentioned before, the precise ratio of $L^{1/4} g_s^{-4/3}$ comes from an additional technical step in going from the background Eq. (96) to, for example, the spectral curve $\mathcal{S}$ of pure topological

gravity or JT gravity, where we zoom in on the edge of the eigenvalue spectrum while taking $L \to \infty$.

## 5  Discussion and Outlook

In this paper we have laid out an arc spanning all the way from the theory of quantum chaos to that of 2d quantum gravity. The guiding principle, from the gravity perspective is the (doubly) non-perturbative completion of the semi-classical path integral of JT-gravity. From the perspective of quantum chaos, the relevant contributions determine the hyper-fine structure of the spectrum of energy eigenstates, which are quasi regularly ordered with an average spacing of $e^{-S_0}$ (see Fig. 2). In quantum chaotic theories this spectral structure is universal and determined by a symmetry principle: the breaking and restoration of causal symmetry, [18], as described by a remarkably simple low-dimensional matrix theory, the fMT introduced in Section 2.1.

The one line summary of our story is that all relevant players in the two-dimensional gravitational framework — Kodaira-Spencer field theory, a system of D-branes introduced into a six-dimensional Calabi-Yau manifold, and the SYK model (at the spectral edge) as a putative boundary theory — map onto that fMT. This proves that they all faithfully describe the universal ergodic phase of quantum chaos – a finding not easily established otherwise. It also establishes their quantitative equivalence in the long time limit. The setting is minimal in that further reduction, such as the perturbative representation of Kodaira-Spencer field theory in terms of the JT gravitational path integral, looses information on the hyperfine structure of the spectrum.

The mechanism behind the reduction to fMT is the fast ergodization of systems with quantum chaotic dynamics. It implies the efficient entangling of all states in a Hilbert space of high 'color' dimension, to the effect that long time correlations are described by a universal effective theory whose low 'flavor' dimension is determined by the number of probes (or the order of correlation functions) into the chaotic phase. This color-flavor duality is universally observed in ergodic quantum chaos and, as we show in this paper, the effective theories of two-dimensional gravity are no exception. The relationship between large-$L$ 'color' matrix model and finite-size 'flavor' matrix models, in particular of the Kontsevich had been appreciated before in the topological- and minimal-string context [31, 34], but it is very illuminating to see that it is in fact an expression of the universality of quantum chaotic correlations even in the gravitational context, as we have established in this work.

In Section 3 we described the above reduction to flavor theory for Kodaira-Spencer (KS) field theory, with brane and anti-brane insertions assuming the role of spectral probes. This setting led to a beautiful geometric view of the analytic structure of the spectrum of 2D quantum gravity: KS theory is defined on the multi-sheeted spectral curve of JT gravity with a branch cut describing the coarse grained spectral density. Its perturbative expansion in $e^{-S_0}$ equals the 'mini-universe expansion' of JT gravity, and at the same time reveals long ranged (on scales of the microstate spacing) correlations in the spectrum. However, it takes a computation non-perturbative in $e^{S_0}$, in the presence of probe branes, to reveal the micro structure of states as poles, rather than elements of a continuous cut. Within the fMT framework, the discontinuity across the cut, and its resolution into individual poles at hyperfine scales are associated to the breaking and restoration of causal symmetry, respectively. It would be interesting to connect this to recent work on the discrete spectrum of a putative non-perturbative completion of JT gravity [78].[17] In Section 4 we looked at these principles from an even further

---

[17]That work also noted the analogy to (classical) D-brane positions being 'smeared' into a continuous spectral density. It would be interesting to find the analog of the Fredholm determinant used by [78] directly in topological

expanded geometric perspective. Starting from the parent geometry of a six-dimensional CY manifold, we showed how it emerges from the world-volume theory of non-compact branes in a closed-string background obtained by dissolving a stack of compact branes into the geometry. In this setting, causal symmetry arises as the world-volume symmetry of the flavor branes. The breaking and restoration of causal symmetry indicates a change in the analytical structure of the target space perceived by the flavor branes: at perturbative level in $e^{S_0}$ the target space presents branch cuts, which are resolved into poles at the fully non-perturbative level. The semi-classical brane partition function expanded around the saddle points presents branch points, while the full quantum result is an entire function of the energy, that is target-space coordiante (e.g. the Airy function at the topological point).

In theories of quantum chaos, the restoration of causal symmetry in the long time limit reflects the hyperfine structure of the chaotic spectrum: levels repel and ultimately 'crystallize' into an approximately evenly spaced structure. Our work shows how the same phenomenon occurs in the gravitational framework, but now can be given a geometric interpretation, as outlined above. We note that story is reminiscent of what happens in the study of the target space geometry of minimal string theory [34] The fact that different effective descriptions of two-dimensional gravity 'want' to contract to fMT at large time scales demonstrates the prevalence of the ergodic quantum chaotic phase in this context. It also makes it tempting to speculate about generalization to other holographic settings, including in higher dimensions. However, the general idea of color-flavor duality extends to the description of time scales shorter than the times required to reach ergodicity, and even to chaotic systems which never reach an ergodic limit. In such cases, flavor theories are promoted to matrix field theories (as opposed to single matrix integrals). Ergodicity is reached *iff* beyond a certain Thouless time scale the inhomogeneous mode spectrum of these theories is frozen out, and a single mean field flavor zero mode governs the field integral (cf. [19, 43] for an execution of this program for the SYK model). One may wonder if a similar reduction may take place in higher-dimensional holographic theories, despite the apparent lack of a precisely defined universe field theory. It would seem promising to try and understand to what extent KS theory can define the ergodic core of universe field theory for higher-dimensional holography, perhaps in an approximate sense.

## Acknowledgements

We would like to thank Jan de Boer, Ricardo Espíndola, Marcos Mariño, Bahman Najian, Pranjal Nayak, Samson Shatashvili and Marcel Vonk for enlightening discussions on the topics of this paper.

**Funding information**  This work has been supported in part by the Fonds National Suisse de la Recherche Scientifique (Schweizerischer Nationalfonds zur Förderung der wissenschaftlichen Forschung) through Project Grants 200020_182513 and the NCCR 51NF40-141869 The Mathematics of Physics (SwissMAP), and the Delta ITP consortium, a program of the Netherlands Organisation for Scientific Research (NWO) that is funded by the Dutch Ministry of Education, Culture and Science (OCW). AA was supported by the Deutsche Forschungsgemeinschaft (DFG) Projektnummer 277101999 TRR 183 (project A03). BP and JvdH were supported by the European Research Council under the European Unions Seventh Framework Programme (FP7/2007-2013), ERC Grant agreement ADG 834878. We would like to thank the Institute des Etudes Scientifiques de Cargèse, and the SwissMAP research station at Les Diablerets for hospitality while this work was in progress.

---

string / KS theory.

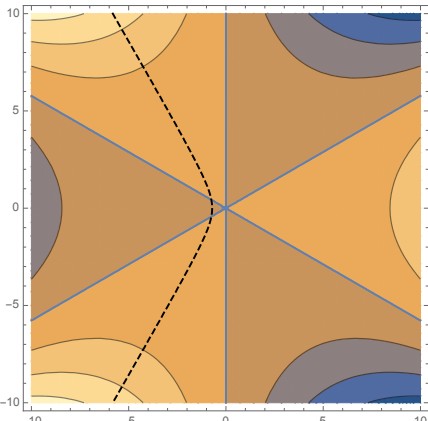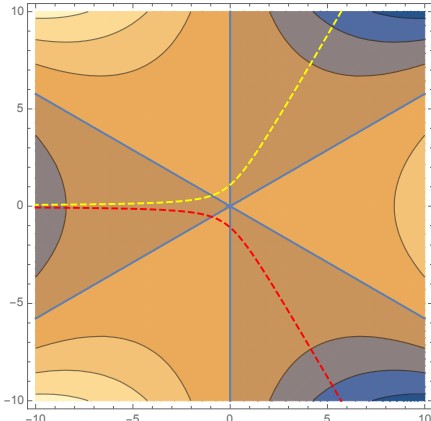

Figure 5: Real part of the Airy potential $\frac{y^3}{3}$. The blue shaded areas are regions where the real part is negative. Left, black striped: integration contour $\mathcal{C}$ for the brane insertions. Right, red and yellow striped: two distinct choices of $\mathcal{C}'_{\pm}$ for the anti-brane insertion.

## A Steepest descent contours for the fMT

In this appendix we study the steepest descent contours for the eigenvalue integrals of the fMT. Naively, looking at the transformations $\psi(x) = \int_{\mathcal{C}} dy\, e^{xy} \widehat{\psi}(y)$ and $\psi^{\dagger}(x) = \int_{\mathcal{C}'} dy\, e^{-xy} \widehat{\psi^{\dagger}}(y)$ for the brane and anti-brane vertex operators, one might expect that $\mathcal{C}$ will be a contour along the imaginary axis, and $\mathcal{C}'$ along the real axis. This is also what one expects based on the color-flavor duality in the finite $L$ matrix theory. However, in the double-scaling limit we have to make sure that the contours $\mathcal{C}, \mathcal{C}'$ go to infinity in a region where the potential grows, to ensure that the flavor matrix integral converges.

Let us first consider the the case of a single brane insertion

$$\langle \psi(x_i) \rangle_{\mathsf{KS}} = \int_{\mathcal{C}} \frac{dy}{\sqrt{\lambda}}\, e^{\frac{1}{\lambda}x_i y - \frac{1}{\lambda}\Gamma_0(y)(1+\mathcal{O}(\lambda))}\,. \tag{A.1}$$

For the purposes of our analysis, we have kept only the leading order term as $\lambda \to 0$. In the case of the Airy curve the potential is given by

$$\Gamma_0(y) = -\langle \widehat{\Phi}(y) \rangle_0 = \int x(y)dy = \frac{y^3}{3}\,, \tag{A.2}$$

and so the integral in Eq. (A.1) becomes the well-known integral representation of the Airy function. The real part of $y^3$ is positive in three wedges of the complex $y$-plane, and there are two independent non-trivial choices of contour, defining the 'Airy' and the 'Bairy' function, respectively. The Airy contour $\mathcal{C}$ can be chosen along the imaginary axis, as long as it remains in the left half-plane (see the black striped contour in Fig. 5.).

Similarly, for a single anti-brane insertion we obtain the integral

$$\langle \psi^{\dagger}(x_i) \rangle_{\mathsf{KS}} = \int_{\mathcal{C}'} \frac{dy}{\sqrt{\lambda}}\, e^{-\frac{1}{\lambda}x_i y + \frac{1}{\lambda}\Gamma_0(y)(1+\mathcal{O}(\lambda))}\,. \tag{A.3}$$

In the Airy case, we now see that the naive integral along the real axis diverges, since $y^3$ blows up as $\mathrm{Re}(y) \to \infty$. The integration contour may start on the negative real axis, but then it should enter into one of the two asymptotic regions $\frac{\pi}{6} < |\mathrm{Arg}(y)| < \frac{\pi}{2}$, such as the red $\mathcal{C}'_{-}$ or yellow $\mathcal{C}'_{+}$ striped contours in Fig. 5. Of course, another valid option would be to integrate

parallel to the imaginary axis in the right half-plane, but by Cauchy this contour can always be deformed to a linear combination of the red and yellow contours.

We can repeat the analysis for the JT spectral curve. In this case, the spectral curve equation is solved by $x(y) = \arcsin^2(y)$, and the leading order potential is

$$\Gamma_0(y) = -2y + \sqrt{1-y^2}\arcsin y + y\arcsin^2 y. \qquad (A.4)$$

Its real part has been plotted in Fig. 6. As one can see, for small $y$, the real part is very similar to the Airy case, but its large $y$ behavior is different. In particular, for large $y$, the complex $y$-plane is divided into only two regions: in the left half-plane, the real part is positive for large enough $\mathrm{Re}(y)$, while in the right half-plane it becomes negative.[18] Moreover, there is a branch cut from the analytic continuation of the $\arcsin(y)$ on two pieces of the real axis $(-\infty, -1] \cup [1, \infty)$. This means that for the brane insertions Eq. (A.1), we can choose the integration contour $\mathcal{C}$ to be homotopic to the Airy contour, as long as it passes through the real axis in the interval $(-1, 1)$ (see also Ref. [59]).

However, for the anti-brane Eq. (A.3), the contours $\mathcal{C}'_\pm$ no longer give rise to a convergent integral, because the integral parallel to the negative real axis grows exponentially for sufficiently negative $\mathrm{Re}(y)$. One possible solution is to integrate y parallel to the imaginary axis, shifted into the positive half-plane. However, if we want to make contact with some finite $L$ flavor integral pre-double scaling, this contour choice should be excluded, for the integration contour in the anti-brane sector pre-double scaling is parallel to the real axis (see, for example, the analysis in Appendix A of Ref. [19]).[19] The resolution of this apparent tension is to take into account the branched structure of the potential $\Gamma_0(y)$. The plot above was generated using the principal branch of $\arcsin(y) = -i\log(iy + \sqrt{1-y^2})$, but there are infinitely many branches, related by $\arcsin(y) + 2\pi k$, for $k \in \mathbb{Z}$. As an example, we have plotted the real part of $\Gamma_0(y)$ taking the branch $k = -1$: We see that a contour which passes through the branch cut onto the next sheet enters a sheet where the potential grows again. We will select precisely such a contour for the anti-brane integrals $y_i$.

Having discussed the asymptotic behavior of the integration contours, we want to deform them into steepest descent (or stationary phase) contours passing through the saddle points.

---

[18]The lines where the sign of the potential changes are indicated in light blue.

[19]On a technical level, the integration parallel to the real axis ensures that the Hubbard-Stratonovich transform (necessary in the color-flavor duality) is well-defined [39].

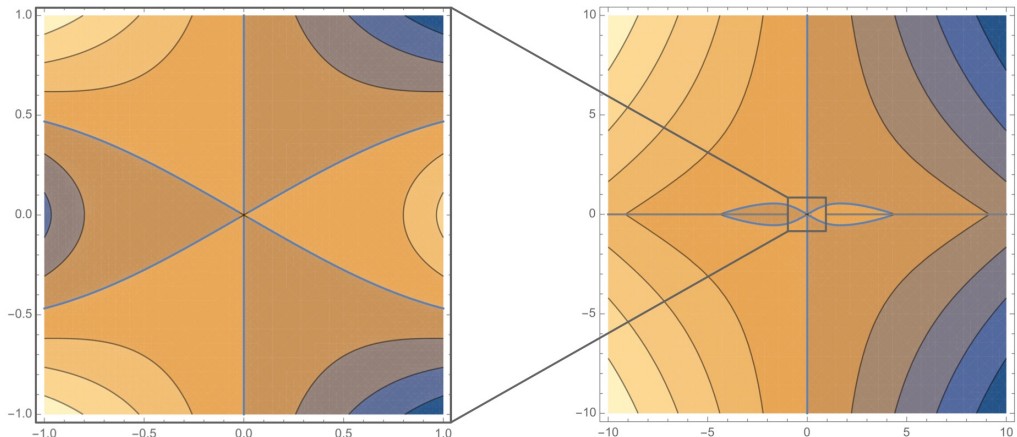

Figure 6: Real part of the JT potential. Close to the origin, the potential looks like Airy potential, while the behavior at infinity is different: the potential is positive and growing for $\mathrm{Re}(y) \ll 0$, while it is negative for $\mathrm{Re}(y) \gg 0$.

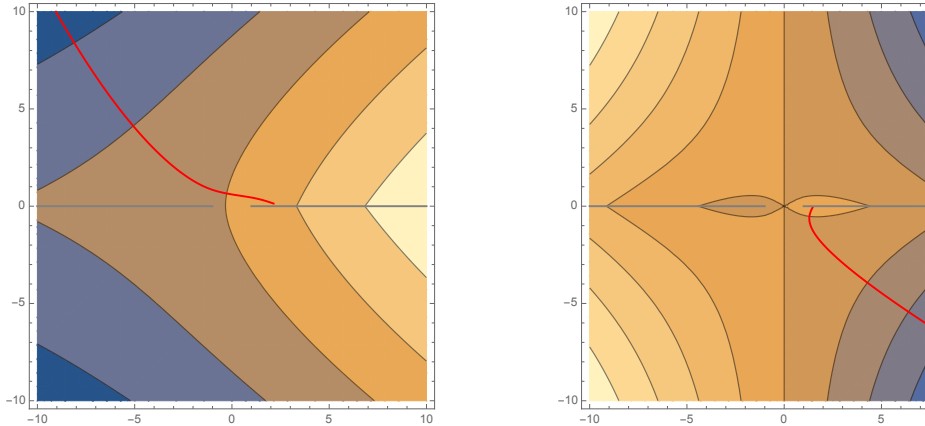

Figure 7: Right: principal branch of $\Gamma_0(y)$. Left: the $k = -1$ sheet of $\Gamma_0(y)$. In red a steepest descent contour that goes to infinity in different sheets of the multiple cover.

Let us first briefly describe the Airy case of a single brane insertion. Varying the action in Eq. (A.1) we find two saddle points, $y^{\pm} = \pm\sqrt{x}$. If we position the brane slightly above the negative $x$-axis in the spectral $x$-plane, $x = -E + i\eta$, the saddle points $y^{\pm}$ lie on opposite sides of the imaginary $y$ axis. The steepest descent (and ascent) contours through these saddle points are plotted in figure 8 (left). The original black dashed Airy contour can be deformed to the sum of the red and blue descent contours, so both saddle points contribute to the integral. However, the contribution of $y^{+}$ is exponentially suppressed compared to that of $y^{-}$. On the other hand, if we position the brane slightly below the negative $x$-axis, $x = -E - i\eta$, the dominant contribution will come from $y^{+}$ instead of $y^{-}$, see Fig. 8 (right). This is an example of a Stokes' phenomenon: the relative dominance between the saddle points is exchanged when we cross the *anti-Stokes line* on the branch cut of $\sqrt{x}$.

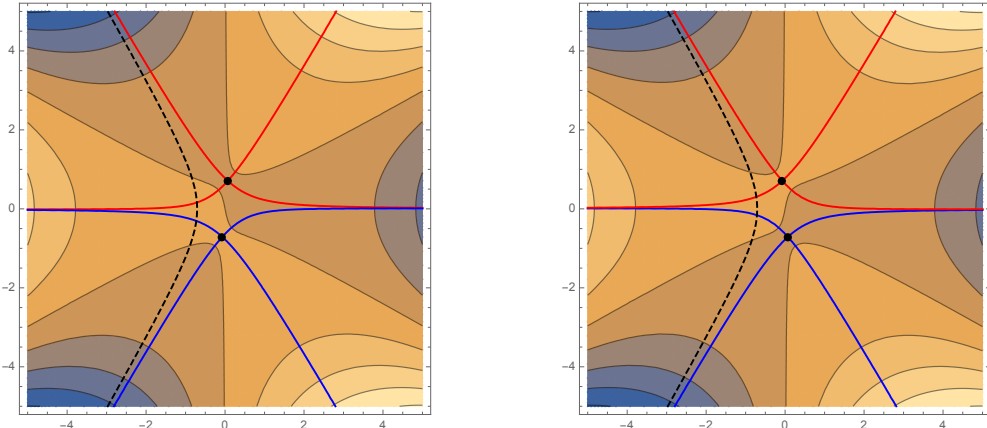

Figure 8: Background: real part of $xy - \frac{y^3}{3}$. Shaded areas are $< 0$, bright areas $> 0$. Black dots are the saddle points. The steepest descent+ascent contours passing through them are drawn in red and blue. Left: $x = -E + i\eta$, right: $x = -E - i\eta$.

For the anti-brane, the original integration contour can only be deformed to one of the steepest descent contours, passing through only one saddle. The red dashed contour in Fig. 5 is deformable to the steepest descent contour passing through $y^{-}$, while the yellow dashed contour is deformable to the steepest descent path through $y^{+}$. This is another example of a Stokes phenomenon: which saddles (cease to) contribute depends on the argument of the external parameter $x$.

Let us now turn to the case of JT gravity. The saddle points are located at

$$y^{\pm} = \pm \sin\sqrt{x}. \tag{A.5}$$

The steepest descent (and ascent) contours through the saddles have been plotted in Fig. 9, for $x = -E + i\eta$ (left) and $x = -E - i\eta$ (right). The black dotted line is the original integration contour for the brane insertions. It can be deformed to the sum of the red and blue steepest descent contours, as these go to the same asymptotic infinity on the next sheet. So both saddle points contribute, and which saddle dominates is determined by the $\pm i\eta$ prescription exactly as in the Airy case. For the anti-brane, one should follow a steepest descent contour that passes through sheet $k = -1$ from an asymptotic infinity, enters the principal branch through the left branch cut and then goes through *only one* of the saddle points. This is again identical to the Stokes' phenomenon of the Airy anti-brane. So we see that the pattern of symmetry breaking precisely follows from the $\pm i\eta$ prescription of the external matrix $X$, in the same way that this was the case for the finite size matrix integrals of [18].

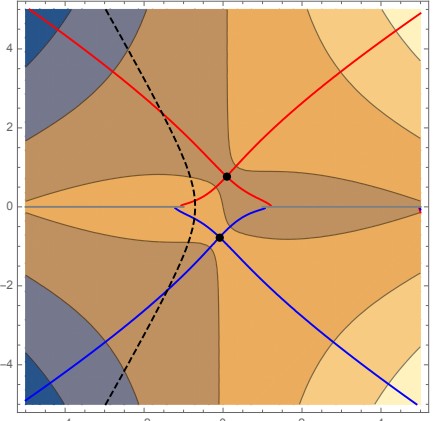 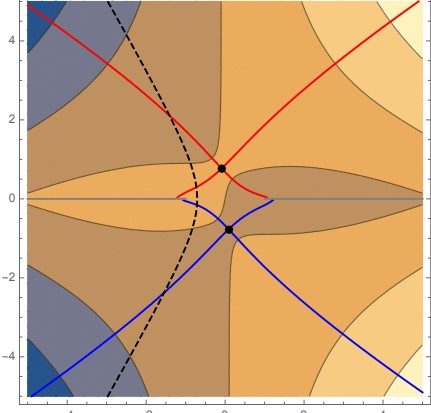

Figure 9: Background: real part of $xy - \Gamma_0(y)$. Shaded areas are $< 0$, bright areas $> 0$. Black dots are the saddle points. The steepest descent+ascent contours passing through them are drawn in red and blue. Left: $x = -E + i\eta$, right: $x = -E - i\eta$.

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
