# Peer review of "Quantum chaos in 2D gravity"

_SciPost Physics, doi:SciPost Phys. 15, 064 (2023)_

## Round 2 · Referee Report · Anonymous · 2022-8-13

Strengths

1. Derivation of a new equation for ratios of n/m determinants in JT gravity as an n+m dimensional super-matrix integral, the action of which can be determined to desired accuracy in the string coupling.
2. Nice discussion on steepest descent contours in appendix A.

Weaknesses

1. The key sections of the paper containing new material (sections 3.2, 3.3, 4.1 and 4.2) are a bit too densely written, whereas the review sections could be shortened.

Report

This paper resolves, in my opinion (and amongst other things) an interesting outstanding technical problem, namely to find a JT gravity version of Kontsevich's matrix Airy function representation of D-brane correlators. In principle it should be possible to derive this from the SSS matrix integral representation of JT gravity, however this has not been done to my knowledge. Instead, the authors derive this equation using a closed universe field theory description of JT gravity. This has one major advantage, namely that this allows the authors to work immediately in the double scaling limit.

The main calculations in this regard are presented in section 3.2 and 3.3. In my opnion, they could be presented in a better way to the reader by shortening the discussions leading up to that (in particular about causal symmetry breaking) and by giving more details and background around the main calculations, in particular from (3.24) to (3.28) and around (3.33). This would make this paper accessible to a wider audience.

The results of section 4 are also interesting. Taking all this into account, this very interesting paper most certainly deserves to be accepted for publication, provided that some minor modifications are made based on my suggestions below.

Requested changes

1. In equation (2.2) I think it is false to say that V(H) and V(A) are identical functions. This works for Gaussian potentials because the Hubbard Stratanovich transformation is simple, but using a generalized Hunnard Stratanovich transformation we are only guaranteed by the results of Guhr and others that P(A) is trace class. This statement is made also in an earlier paper by a subset of the authors, but I do not think that paper contains a derivation either. Unless the authors can prove this statement very explicitly, it can not be made, especially since I think it is false. This is not important for the remainder of the paper, in particular the potential Gamma(A) in (3.29) is correct.
2. In the introduction there are certain (in my opninion) misguided statements about JT gravity and the SSS correspondence to a matrix integral, such as saying that "the correspondence between JT and fMT is limited to early times". Then the authors write footnote 1, which is basically saying the opposite. I do not think it has been shown that JT perturbation theory in genus does not know everything that the double scaled matrix integral knows (in fact, I think it is likely that it does). I would ask the authors to modify statements like this, or at least tone them down, to avoid saying potentially wrong things.
3. At the top of page 3, why are the authors saying low energy scales? Perhaps this is a typo and they mean closeby energies? Or perhaps they have in mind low energies in the sense of Fig. 1, saying that we uberhaupt have a meaningful geometric (non-stringy) description? It would be good to clarify this.

  • validity: top
  • significance: good
  • originality: top
  • clarity: ok
  • formatting: perfect
  • grammar: perfect

Author:  Julian Sonner  on 2022-11-11  [id 3008]

(in reply to Report 1 on 2022-08-13)

The report submitted on 2022-8-13 rightfully points out that the potential of the flavor matrix integral is (for the non-Gaussian case) in general different than the potential of the color matrix integral. The new version rectifies this statement (see below Eq.2.2) and we thank the referee for pointing it out to us. Secondly, the referee has identified some ambiguous or imprecise statements in the Introduction. We believe that the new Introduction is precise and correct in its claims, resolving the issues put forth in the referee report. Lastly, the referee found the review section about causal symmetry breaking in disbalance with the main material in section 3 and 4. The new structure, in which all the review material is collected in a single section (which may be skipped by experts), puts more focus on the main new material. This should restore the balance between old and new ideas that the referee was looking for.

Attachment:

RefereeReply_avRLBEc.pdf

---

## Round 2 · Referee Report · Anonymous · 2022-8-29

Strengths

The paper develops a quantitative connection between the matrix theory formulation of 2d gravity and theory of quantum chaos, thus unifying very different branches of physics.

Weaknesses

The paper is difficult to read in the sense that it lack transparency. It describes connections between many different theories (and quantities), but the overall picture is very difficult to grasp. One example is a very complex scheme in Fig. 2 which outlines connections between different parts. To fully appreciate the paper requires a very significant background from the reader.

Report

This is without a doubt a very interesting paper which develops a connection between the matrix model description of 2d quantum gravity with the matrix model which appears in the description of quantum chaos, i.e. that yields universal aspects of spectral statistics found in chaotic systems. The underlying idea goes back to Ref. [22]. Current paper provides more context and tries to built on the mathematical relations to develop a geometric picture. Namely, the matrix model of interest has an interpretation in terms of a limit of certain sigma model on 6d manifolds. Apparently the paper aims to recognize corresponding geometric structures on the side of 2d gravity.

The paper is written by experts in the field, which assures that technical statements are correct. Yet, the overall picture outlined in the paper is not clear, that is despite the authors attempts. I believe this paper should be published, after a revision which would introduce a more approachable explanation of the results, which would not rely on the extensive prior knowledge.

Requested changes

I believe the introduction and(or) conclusions should be rewritten to explain the overall picture. Which mathematical statements in the paper are new and which have been known before? Which quantitative relations (i.e. equality between quantities in different theories, aka "dualities") observed in this paper are new and which ones were known before? What is the overall conceptual picture? What is the role of 3D Chern-Simons theory and string theory (in how many dimensions?) in describing 2d quantum gravity? I believe for paper to be approachable beyond a very limited group of experts on both recent developments in 2d gravity and those, knowledgeable in earlier works on Kodaira-Spencer field theory and topological strings, it requires a down to earth explanation of the overall picture.

  • validity: good
  • significance: good
  • originality: high
  • clarity: low
  • formatting: excellent
  • grammar: excellent

Author:  Julian Sonner  on 2022-11-11  [id 3007]

(in reply to Report 2 on 2022-08-29)

In this report, submitted on 2022-8-29, the referee stresses the need to revise the Introduction such that it outlines the main ideas and makes the work accessible to a wider audience. We believe that the new Introduction does precisely this: it explains the motivation to connect JT gravity to the field of quantum chaos, using the tools and intuition from (topological) string theory, after which it presents the main new equations derived in Section 3, followed by an open string interpretation and finally its relevance to SYK. The old 'metro map' figure is gone and has been replaced be a more coherent overview diagram (Figure 1).

Furthermore, the referee points out that to fully appreciate the connections made in the article, the reader needs to be an expert in both fields. In order to increase readability we have made changes throughout highlighting the steps taken and ideas required. We now introduce and collect background material in Section 2. The other ingredients are presented in small doses as one goes along. We believe that this way the reader will stay with us without being intimidated by too much review content. In combination with a better explanation of the overall picture, the current article increases the accessibility to a wider audience that the referee sought for.

Attachment:

RefereeReply.pdf

---

## Round 3 · Referee Report · Anonymous (Referee 2) · 2022-11-14

Report

In the revised version the authors have substantially rewritten the introduction to better explain their results. This change was necessary; the paper assumes knowledge of the Kodaira-Spenser theory. Current version is much more accessible to a general reader, and conveys the logic of the main results quite well. I think the paper is ready to be accepted for publication.

---

## Round 3 · Referee Report · Anonymous (Referee 1) · 2022-12-11

Report

With these revisions the authors adressed my minor points of concern. Combining all background material in section 2 was an improvement in my opinion, and makes the paper more readable. I recommend the paper for publication.

---

## Round 3 · Author Response

Dear editor-in-charge,

We thank the referees for their positive review and constructive criticism of our paper `Quantum chaos in 2D gravity', submitted to SciPost Physics on 2022-07-01.
The referees clearly showed an interest in our paper and a sharp eye for its possible improvements. After reading the referee reports and Editorial Recommendation, we have incorporated the suggestions made therein, resulting in this revised version of the article submission. We will now list the changes made compared to version 1, in response to both reports.

---

## Round 3 · List of Changes

The main revision to the article is a rewriting of the Introduction. As both referees have independently noted, the original introduction lacked clarity of presentation as well as a clear statement of the new results presented in this article. We believe that the new Introduction is more structured and unambiguously states the main aim, ideas and equations derived in the body of the work. Moreover, the original overview diagram (Figure 2 in the first version) has been replaced by a slimmed down diagram which highlights the principal connections made in the paper.

The other structural revision has been to combine all the reviewed material and necessary prior knowledge in a separate section, aptly titled 'Setting the scene'. The new material following this section has been made self-contained and more flowing. Moreover, some extra details of the computation are added where necessary. Lastly, a few minor typos were fixed and some references were added.

---

## Editorial Decision

published